# Hallmarks of a genomically distinct subclass of head and neck cancer

Tara Muijlwijk [1,2,3], Irene H. Nauta[1,2], Anabel van der Lee[1,2,3], Kari J. T. Grünewald[1,2], Arjen Brink[1,2], Sonja H. Ganzevles[1,2,3], Robert J. Baatenburg de Jong[4], Lilit Atanesyan [5], Suvi Savola [5], Mark A. van de Wiel [6], Laura A. N. Peferoen[2,7,8], Elisabeth Bloemena [2,7,8], Rieneke van de Ven[1,2,3], C. René Leemans [1,2], Jos B. Poell [1,2,9] ✉ & Ruud H. Brakenhoff [1,2,9] ✉

Cancer is caused by an accumulation of somatic mutations and copy number alterations (CNAs). Besides mutations, these copy number changes are key characteristics of cancer development. Nonetheless, some tumors show hardly any CNAs, a remarkable phenomenon in oncogenesis. Head and neck squamous cell carcinomas (HNSCCs) arise by either exposure to carcinogens, or infection with the human papillomavirus (HPV). HPV-negative HNSCCs are generally characterized by many CNAs and frequent mutations in *CDKN2A*, *TP53*, *FAT1*, and *NOTCH1*. Here, we present the hallmarks of the distinct subgroup of HPV-negative HNSCC with no or few CNAs (CNA-quiet) by genetic profiling of 802 oral cavity squamous cell carcinomas (OCSCCs). In total, 73 OCSCC (9.1%) are classified as CNA-quiet and 729 as CNA-other. The CNA-quiet group is characterized by wild-type *TP53*, frequent *CASP8* and *HRAS* mutations, and a less immunosuppressed tumor immune microenvironment with lower density of regulatory T cells. Patients with CNA-quiet OCSCC are older, more often women, less frequently current smokers, and have a better 5-year overall survival compared to CNA-other OCSCC. This study demonstrates that CNA-quiet OCSCC should be considered as a distinct, clinically relevant subclass. Given the clinical characteristics, the patient group with these tumors will rapidly increase in the aging population.

Head and neck squamous cell carcinoma (HNSCC) is the seventh most common cancer worldwide and has a mortality rate of approximately 50% with annually 890,000 new cases and 450,000 deaths[1]. HNSCC arise in the mucosal linings of the upper aerodigestive tract and are most frequently located in the oral cavity, hypopharynx, nasopharynx, larynx, or oropharynx. Among HNSCC, carcinomas of the oral cavity (OCSCC) are most prevalent with an increase in incidence over the years[1]. Risk factors for nasopharyngeal cancers, most prevalent in East

[1]Amsterdam UMC, location Vrije Universiteit Amsterdam, Otolaryngology / Head and Neck Surgery, Amsterdam, The Netherlands. [2]Cancer Center Amsterdam, Cancer Biology and Immunology, Amsterdam, The Netherlands. [3]Amsterdam Institute for Infection and Immunity, Cancer Immunology, Amsterdam, The Netherlands. [4]Erasmus University Medical Center, Otorhinolaryngology / Head and Neck Surgery, Rotterdam, Netherlands. [5]MRC Holland, Oncogenetics, Amsterdam, The Netherlands. [6]Amsterdam UMC, Epidemiology & Data Science, Amsterdam Public Health Research Institute, Amsterdam, The Netherlands. [7]Amsterdam UMC, location Vrije Universiteit Amsterdam, Pathology, Amsterdam, The Netherlands. [8]Academic Center for Dentistry, Maxillofacial Surgery/ Oral Pathology, Amsterdam, The Netherlands. [9]These authors jointly supervised this work: Jos B. Poell, Ruud H. Brakenhoff. ✉e-mail: j.poell@amsterdamumc.nl; rh.brakenhoff@amsterdamumc.nl

and Southeast Asia, are tobacco smoking and Epstein-Barr virus (EBV) infection[2]. Tumors of the oral cavity, hypopharynx, and larynx, which are more commonly found in Western countries, are predominantly carcinogen-induced, with smoking and excessive alcohol consumption as classical risk factors. In contrast, oropharyngeal squamous cell carcinomas (OPSCCs) are increasingly caused by human papillomavirus (HPV) infection. HPV-positive and HPV-negative OPSCC are considered as separate disease entities due to the differences in etiology, molecular characteristics, clinical presentation and prognosis. HPV-positive OPSCCs have a substantially more favorable prognosis than HPV-negative tumors[3–5].

HPV-negative HNSCC are generally characterized by many copy number alterations (CNAs) as well as frequent mutations in the tumor suppressor gene *TP53*[4]. *TP53* is located at chromosome 17p13 and encodes the tumor suppressor p53 that responds to cellular stress signals such as DNA damage[4,6]. Strikingly, Smeets et al.[7] identified an HPV-negative HNSCC subclass of tumors with few or no chromosomal gains and losses. These tumors were characterized by wild-type (wt) *TP53*[7]. The cancer genome atlas (TCGA) network also reported on the existence of this group with low numbers of chromosomal aberrations and further noticed that those tumors, besides *wtTP53*, were enriched for inactivating mutations of caspase 8 (*CASP8*) and activating mutations of *HRAS*[8].

Of note, these studies met certain limitations. The first and most problematic issue is that both studies did not account for the bias introduced by high stromal contamination. Specifically, a low fraction of tumor cells (purity) in the sample may completely obscure the tumor-specific copy number signal, resulting in a lack of power to call CNAs. This may consequently result in underestimation of CNAs.

Secondly, in both studies clustering methods were employed to define the CNA-quiet group instead of applying a quantitative measure, such as the fraction genome altered (FGA), which hampers interpretability, reproducibility across cohorts, and application of the classification to single samples. Lastly, until now, this particular subclass is only studied in small[7] or heterogeneous cohorts that also include HPV-positive tumors[8]. In order to translate stratification of the CNA-quiet tumors to other cohorts and to the clinic, a different method is warranted. Precise definition and comprehensive clinical, histological, molecular, and immunological characterization of CNA-quiet tumors in a large homogeneous cohort is lacking at present.

In this study, we define CNA-quiet head and neck cancers through in-depth analysis of 409 HPV-negative tumors from the TCGA, validate the findings in a multicenter cohort of 802 OCSCC, and investigate clinical, histological, and immunological associations. Here, we set the standard to define and characterize this clinically, histologically, and immunologically distinct subclass of head and neck cancers (Fig. 1).

## Results

### Defining CNA-quiet tumors in the cancer genome atlas cohort

In the search for a quantitative measure for CNA classification, the HPV-negative HNSCC TCGA cohort was utilized. First, we explored the FGA provided by cBioPortal[9] from 409 HPV-negative HNSCC (Supplementary Fig. 1 for TCGA cohort filtering). Realizing that tumor percentage may impact FGA values, we analyzed this association in TCGA data and observed that the provided FGA correlated strongly with the percentage of tumor cells (purity, Fig. 2a, $r = 0.52$, $p < .001$). We noted that this FGA metric is likely biased by the fact that CNA calling by the TCGA was based on genomic segment signals that are

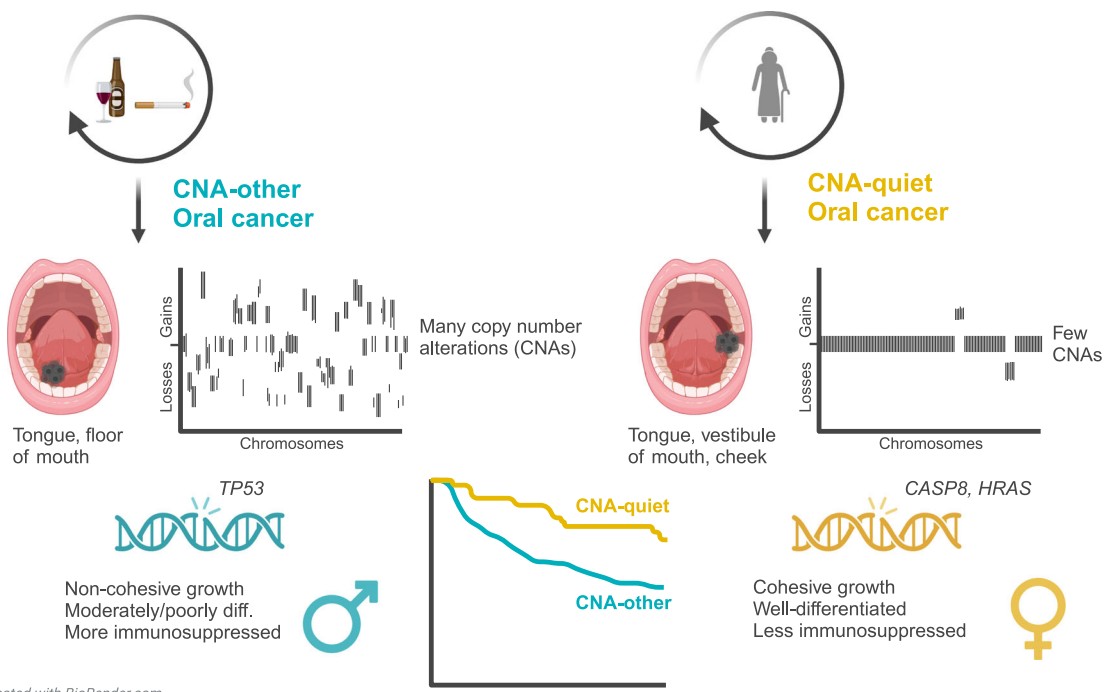

# Hallmarks of a Genomically Distinct Subclass of Head and Neck Cancer

**Fig. 1 | Hallmarks of a genomically distinct subclass of head and neck cancer.** Characterization of the copy number alteration (CNA)-quiet subclass. CNA-quiet oral cancers are predominantly wild-type *TP53*, frequently harbor *CASP8* and *HRAS* mutations, exhibit a cohesive growth pattern, are often well-differentiated, and have a less immunosuppressed tumor immune microenvironment. Both oral cancer subclasses are most commonly located on the tongue. CNA-quiet tumors are also frequently found in the vestibule of mouth and cheek mucosa, while CNA-other cancers are more commonly located on the floor of mouth. The CNA-quiet group is enriched for females without tobacco use, less or no alcohol use, and a higher age at diagnosis. Patients with CNA-quiet oral cancers demonstrate a more favorable 5-year overall survival compared to CNA-other oral cancers. Figure 1 is created with BioRender.com released under a Creative Commons Attribution-NonCommercial-NoDerivs 4.0 International license.

not corrected for stromal contamination. Low FGA values may therefore be the result of a lack of power to call CNAs in low purity tumors, resulting in an underestimation of the FGA and thus leading to false CNA-quiet tumors (Fig. 2b). To correct for stromal contamination, we reassessed tumor purity of the samples. Consensus fits for tumor purity and ploidy were derived using R packages ACE[10], ABSOLUTE[11] and a combination of both. To compare these methods, cases with a presumed FGA lower than 0.10 were selected ($n = 87$) and analyzed by the three methods (Supplementary Fig. 2). The best fit was selected on the basis of the likelihood of copy number profiles and variant copies of mutated genes. Reasonably comparable or equally likely fits were accepted as well. The combination of ACE and ABSOLUTE was considered the most optimal fitting strategy to determine tumor purity (Supplementary Fig. 2b, c). Using these optimal purity and ploidy estimates, we recalculated segment values. Indeed, this recalculation reverted the confounding effect of tumor purity on FGA (Fig. 2c).

Next, we questioned whether the FGA could be used to define CNA-quiet tumors. We initially divided the tumors in four FGA categories based on the distribution of the recalculated FGA across the TCGA cohort (Fig. 2d). We assessed the presence of *CASP8* and *HRAS* mutations across the four suggested categories, considering prior knowledge of their enrichment in CNA-quiet tumors as reported by the TCGA[8] (Fig. 2e). When taking into account those mutations, both categories with FGA smaller than 0.20 similarly contributed to the enrichment of *HRAS* and *CASP8* mutations (Fig. 2e, f). However, solely using the FGA as determinant for CNA classification revealed that some tumors with a typical CNA-high profile but with many small segments and consequently a low FGA had to be categorized as CNA-quiet (Supplementary Fig. 3). By using the FGA in combination with the number of called segments as cut-off for CNA-quiet tumors, this issue was circumvented (Fig. 2g). We applied this cut-off to the filtered HPV-negative HNSCC TCGA cohort to end up with a group of 70 CNA-quiet tumors with FGA < 0.20 and < 20 called segments and 339 CNA-other tumors with FGA ≥ 0.20 and/or ≥ 20 called segments (Fig. 2h, Supplementary Data 1 for TCGA cohort with recalculated FGA). Of note, CNA-quiet tumors developed almost exclusively (91%) in the oral cavity (Fig. 2i).

## Multicenter validation cohort

To characterize the hallmarks of CNA-quiet tumors in detail, and realizing that these almost exclusively occur as oral cancers, we analyzed a large homogeneous cohort of OCSCC from two centers: Amsterdam UMC and ErasmusMC, with long-term follow-up. We obtained formalin-fixed paraffin-embedded (FFPE) primary tumor specimens from a consecutive cohort of 900 patients treated between 2008 and 2014.

## Multiplex ligation-dependent probe amplification as pre-screening tool

The gold standard to obtain CNA profiles is by performing low coverage whole genome sequencing (lcWGS), but this method is costly and labor-intensive. Therefore we tested and used multiplex ligation-dependent probe amplification (MLPA) as pre-screening. MLPA is applied for a variety of research and clinical applications to assess copy number changes and larger indels[12,13]. For the development of the MLPA probe set, we selected MLPA probes in regions most frequently gained and lost in HNSCC[8]. Next, we performed a validation study in which both MLPA and the gold standard lcWGS were performed on 44 FFPE tumor samples previously analyzed by lcWGS. We concluded that MLPA is a suitable pre-screening method to select CNA-other OCSCC: the positive predictive value of MLPA to correctly assign samples as CNA-other was 100% (Supplementary Fig. 4). MLPA was performed on the entire cohort of 900 OCSCC, which enabled us to identify 365 CNA-other OCSCC (Fig. 3a). Since MLPA does not provide the FGA, lcWGS was required to stratify the remaining 535 samples. These were referred to as CNA-quiet candidates, as they, besides the true CNA-quiet OCSCC, still comprised CNA-other OCSCC for which copy number changes were either outside the region of the selected MLPA probes or not called due to low tumor purity.

## Low coverage whole genome sequencing and target-enrichment sequencing

The 535 CNA-quiet candidates were subjected to lcWGS, which enabled us to further stratify the cohort into CNA-quiet and CNA-other OCSCC (Fig. 3a). Segments were called on the basis of significant deviation compared to a panel of FFPE-derived normal samples[14]. The FGA was calculated by dividing the length of the called segments by the sum of all segment lengths. Both the FGA and the number of called segments were used to stratify the tumors. As we used a different platform for CN analysis than the TCGA (lcWGS versus single nucleotide polymorphisms (SNP) arrays, respectively), we obtained a distinct segment output. The cut-off of 20 called segments, initially determined for the SNP array data, had to be adjusted to our lcWGS platform. The mean number of called segments of TCGA samples was 3.6 times higher compared to the mean number of called segments using lcWGS (48.3 versus 13.4), despite highly congruent genome-wide CNA frequencies. Therefore, 6 called segments were used as a cut-off when using lcWGS data (Supplementary Fig. 5). Using lcWGS, 43 samples were excluded by quality control (QC), another 364 OCSCC were classified as CNA-other, and 128 samples were categorized as CNA-quiet candidates (Fig. 3a). They are still referred to as CNA-quiet candidates since they might be true CNA-quiet OCSCC, but may also still include samples for which no tumor was present in the FFPE sample.

We next performed target-selected deep sequencing of 29 cancer driver genes frequently mutated in head and neck cancer. First, to confirm presence of tumor cells in samples by somatic mutations without any apparent CNAs, and second to provide an overview of the driver events involved in the carcinogenesis of CNA-quiet OCSCC. Although it is possible that tumors lack both any CNAs and driver mutations in these 29 genes, this is only expected to be very rare. Only 1 out of 409 tumors (0.25%) in the TCGA cohort lacked both CNAs and any mutation in a gene represented in our panel. Tumor purity and ploidy were estimated based on copy number data and in case no CNAs were present, combined with mutation data. Tumor purity was further used to calculate the statistical power to call CNAs and somatic variants within a sample. Genetic summaries including power calculations were produced for all samples with targeted deep sequencing data, examples of summary figures for CNA-quiet and CNA-other OCSCCs are displayed in Supplementary Fig. 6a and 6b, respectively. In total, 55 samples had to be excluded based on QC (tumor purity <0.10 or insufficient power to call CNAs and mutations by noise in the data). The remaining 73 samples were classified as CNA-quiet OCSCC with an FGA of less than 0.20 and less than six called segments (Fig. 3a). Within this group, 53 OCSCC contained few CNAs and one or more driver gene mutation; 18 OCSCC lacked any CNAs but had up to ten mutations present; and two tumors had no mutations in the tested driver genes, but only CNAs. Examples of copy number plots from a CNA-quiet and CNA-other tumor are displayed in Fig. 3b and c, respectively. Overall, from the 900 oral cavity SCC specimens, 98 had to be excluded based on QC, and 802 were used for further analysis of which 729 were categorized as CNA-other and 73 as CNA-quiet OCSCC (9.1%, Fig. 3a).

## Multiple CNA-quiet biopsies demonstrates minor CNA heterogeneity

Previously we tested CNA and mutation intratumor heterogeneity in oral cancer by genetic analysis of multiple biopsies[15]. Generally, CNAs and mutations in the cancer driver genes were very comparable, although some heterogeneity was observed. One out of the eleven studied cases was CNA-quiet and it remained CNA-quiet in the two

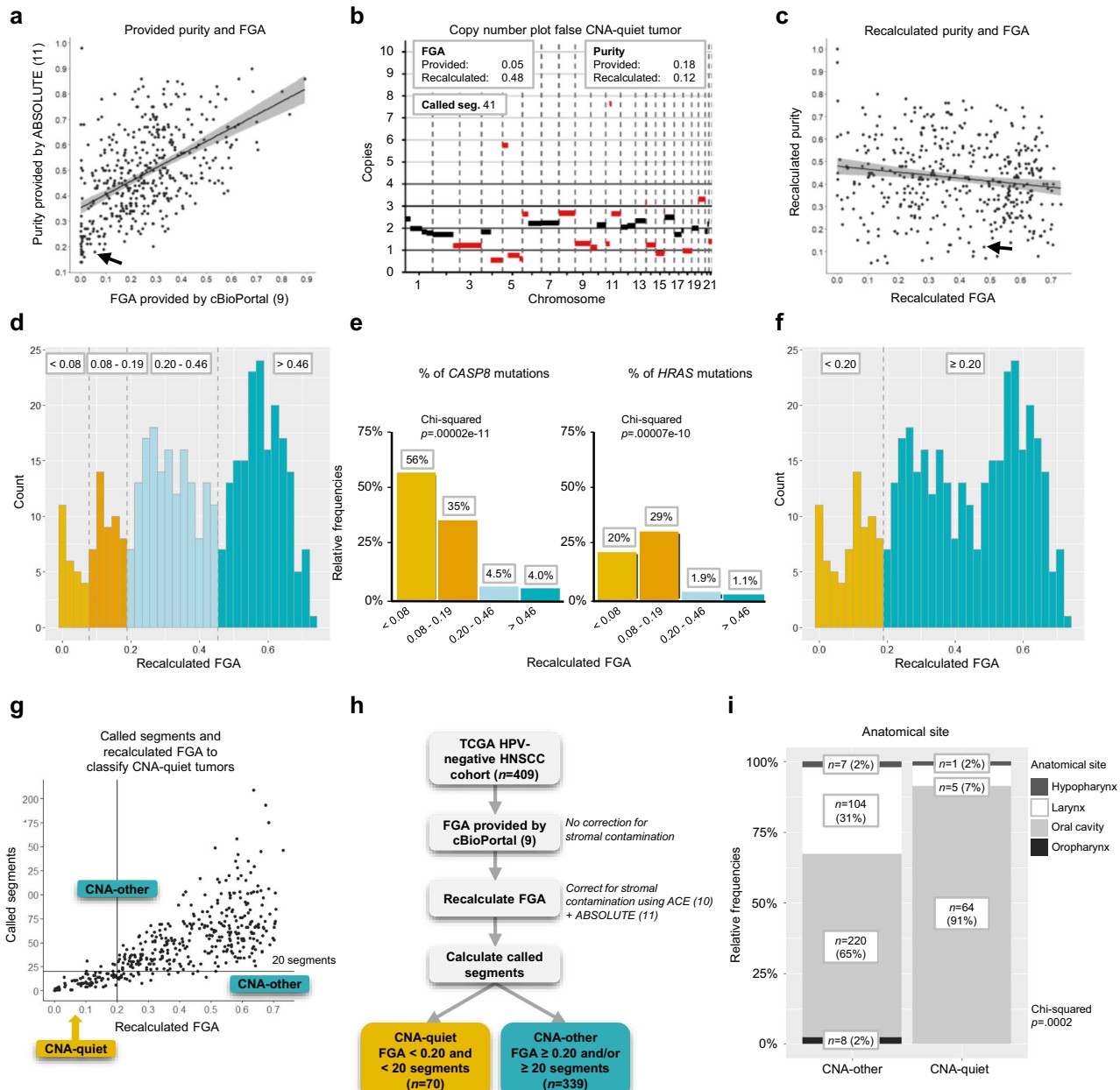

**Fig. 2 | Defining a cut-off to categorize copy number alteration (CNA)-quiet tumors using the cancer genome atlas (TCGA). a** Fraction genome altered (FGA) provided by cBioPortal[9] plotted against purity provided by ABSOLUTE[11] for 409 human papillomavirus (HPV)-negative head and neck squamous cell carcinoma (HNSCC) samples, arrow corresponds to example in (**b**). **b** Example of false copy number alteration (CNA)-quiet tumor with a provided FGA of 0.05 but a recalculated FGA of 0.48 and clearly abundant altered segments. Segments are shown in black and highlighted in red when called. **c** Recalculated FGA plotted against purity with arrow assigning example from (**b**). **d** Distribution of recalculated FGA across 409 HPV-negative HNSCC samples with separation of four groups: FGA < 0.08; FGA between 0.08 and 0.19; FGA between 0.20 and 0.46; and FGA > 0.46. **a**, **c** Black line: linear regression; in gray 95% confidence interval. **e** Percentage of *CASP8* and *HRAS*

mutations across the four CNA groups out of the total HPV-negative HNSCC TCGA cohort. *p*-values were obtained using a two-sided Chi-squared test. **f** Distribution of recalculated FGA across 409 HPV-negative HNSCC samples with separation of two groups: FGA < 0.20 and ≥ 0.20. **g** Called segments (y-axis) and recalculated FGA (x-axis). Tumors with FGA < 0.20 and < 20 called segments were classified as CNA-quiet and tumors with FGA ≥ 0.20 and/or ≥ 20 called segments were classified as CNA-other. **h** The FGA was recalculated, with correcting for stromal contamination, using R packages ACE (Poell et al.[10].) + ABSOLUTE (Carter et al.[11], Supplementary Fig. 2). In total, 70 tumors were assigned as CNA-quiet and 339 as CNA-other. **i** Relative frequency of anatomical site in CNA-quiet and -other groups. *p*-values were obtained using a Chi-squared test. Source data are provided with this paper.

studied biopsies[15]. To extend this group, we tested multiple biopsies of nine CNA-quiet OCSCC. Three examples are shown in Fig. 4 and the remaining cases in Supplementary Fig. 7. Despite some minor changes between biopsies, eight out of nine CNA-quiet OCSCC remained CNA-quiet in all biopsies. One case showed somewhat more CNA heterogeneity, as one out of three biopsies was classified as CNA-other with an FGA just above 0.20 (Fig. 4d).

**CNA-quiet oral cavity SCC have a distinct cancer driver gene profile**

To compare cancer driver gene mutations between the CNA groups, 71 CNA-other OCSCC were randomly selected and subjected to target-enrichment sequencing. To guarantee enough power to call mutations, tumors were selected from all CNA-other OCSCC with a purity higher than 0.20. Summary figures including CNAs and driver gene

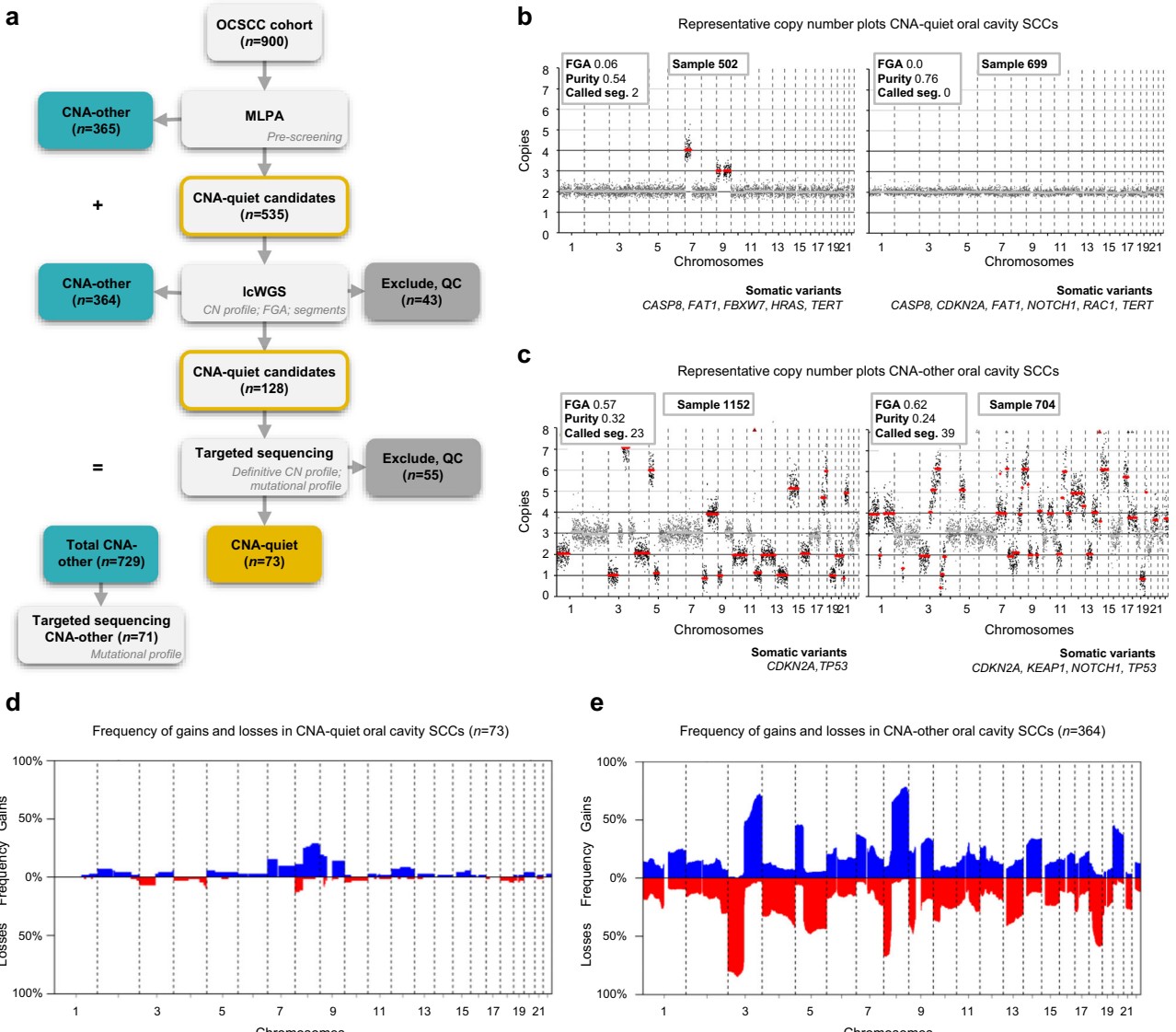

**Fig. 3 | Classification of copy number alteration (CNA)-quiet and CNA-other oral cavity SCC (OCSCC) using a multicenter cohort. a** Flow chart of CNA-quiet classification. First, multiplex ligation-dependent probe amplification (MLPA) was performed on the entire cohort of 900 OCSCC as pre-screening to assign CNA-other OCSCC. CNA-quiet candidates were subjected to low coverage whole genome sequencing (lcWGS) to generate copy number (CN) plots as well as to calculate the fraction genome altered (FGA) by dividing the length of the called segments by the sum of all segment lengths and count the called segments. OCSCC with FGA < 0.20 and < 6 called segments were classified as CNA-quiet candidates, tumors with FGA ≥ 0.20 and/or ≥ 6 called segments were classified as CNA-other, and some

samples had to be excluded for quality control (QC). With the remaining CNA-quiet candidates, targeted sequencing was performed to confirm presence of tumor cells, adjust the definitive CN profile, and for QC. In addition, targeted sequencing was executed to compare the mutational profiles of CNA-quiet and CNA-other OCSCC. **b, c** Representative CN plots of (**b**) CNA-quiet and (**c**) CNA-other OCSCCs. Segments are shown in gray and highlighted in red when called. Tumor purity was determined by the copy number changes using ACE[10] and/or on the basis of variant allele frequency of somatic variants as described in the methods. **d, e** Frequency of gains (in blue) and losses (in red) analyzed by lcWGS for (**d**) 73 CNA-quiet and (**e**) 364 CNA-other OCSCCs. Source data are provided with this paper.

mutations were produced for all specimens (Supplementary Fig. 6). See Supplementary Data 2–4 for the OCSCC cohort with clinical, histological, and mutational details, a list of somatic variants, and the comparison between mutations in CNA-quiet and CNA-other OCSCC, respectively. The high number of telomerase reverse transcriptase (*TERT*) promotor mutations in this cohort is remarkable (Fig. 5a) and corroborates findings from a previous study on *TERT* promoter mutations in OCSCC[16].

As expected, clear differences were found when comparing mutations between the CNA-other and CNA-quiet groups (Fig. 5b). Firstly, in the CNA-other group, *TP53* mutations were evidently more abundant, with 93% of the tumors harboring a mutation in *TP53*, while

only 22% of the CNA-quiet OCSCCs had a mutation in *TP53* (*p* < .001). Missense followed by truncating mutations were the most prevalent *TP53* mutations across both CNA groups; the variant classification did not differ significantly between the groups. In addition, no differences in disruptive or nondisruptive *TP53* alterations were found across the groups[17] (Supplementary Data 5). Secondly, as expected, prevailing somatic variants in CNA-quiet OCSCC included inactivating mutations in *CASP8* (44% versus 13%, *p* = .001) and activating mutations in *HRAS* (19% versus 3%, *p* = .04). Other variants that occurred frequently in CNA-quiet OCSCC were mutations in *PIK3CA* (Supplementary Data 6), *RAC1* and *TERT* promoter. The panel of 29 genes was based on most common mutations in HNSCC in general. We examined the

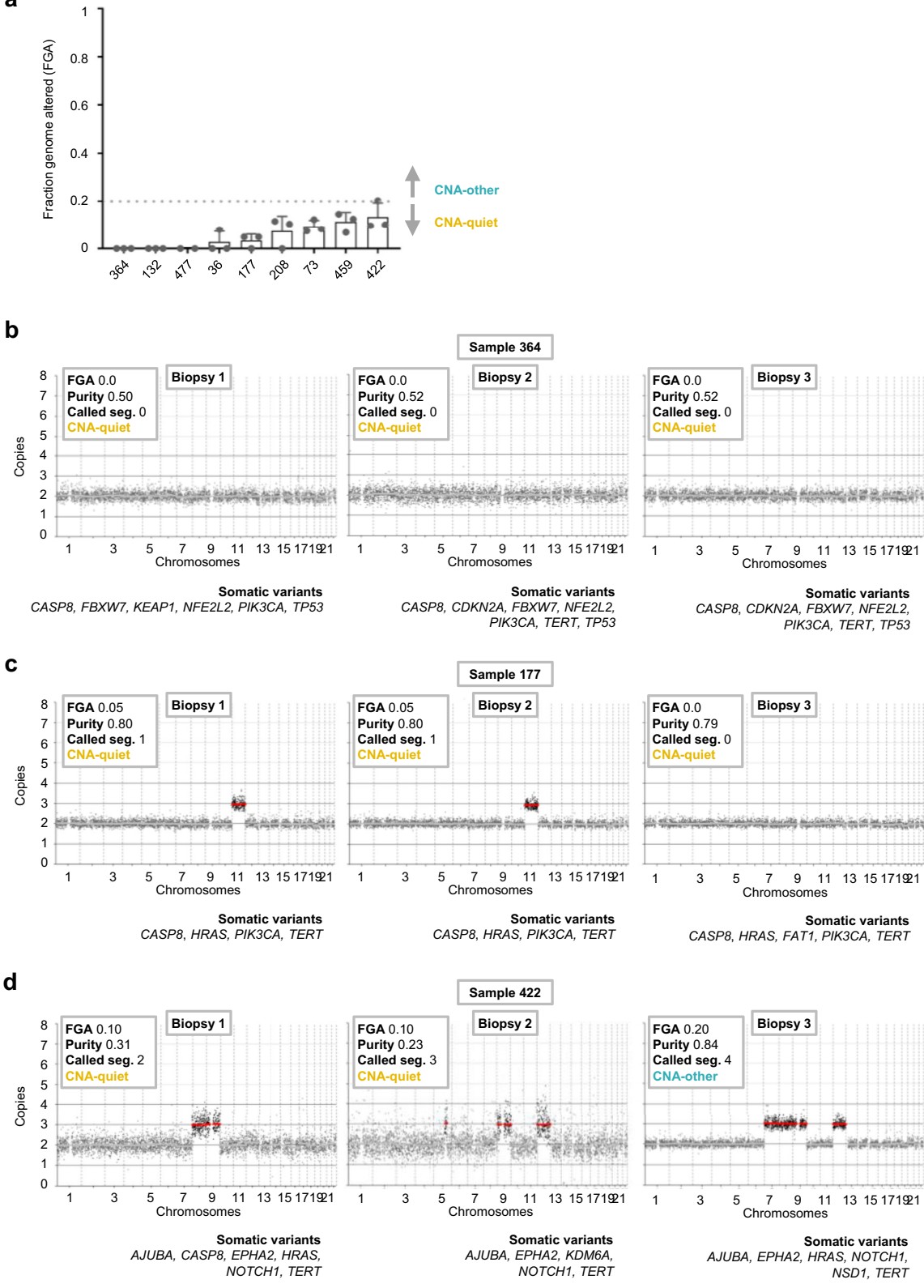

**Fig. 4 | Multiregion sequencing of nine copy number alteration (CNA)-quiet oral cavity SCC (OCSCC) to analyze CNA intratumor heterogeneity. a** Fraction genome altered (FGA, y-axis) across multiple biopsies (*n* = 3 biopsies per tumor, except for sample 477: *n* = 2 biopsies) from *n* = 9 CNA-quiet OCSCCs (x-axis). Data are presented as mean and error bars represent standard deviation. **b−d** Representative copy number plots of sample (**b**) 264, (**c**) 177 and (**d**) 422. Segments are shown in gray and highlighted in red when called, as output from low coverage whole genome sequencing (lcWGS). Source data are provided with this paper.

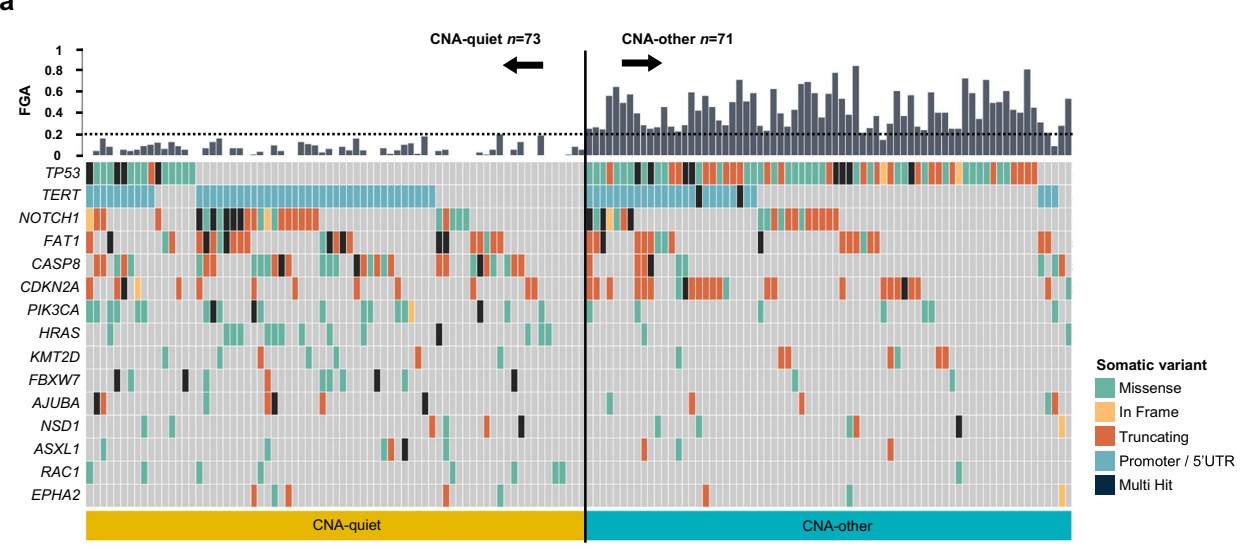

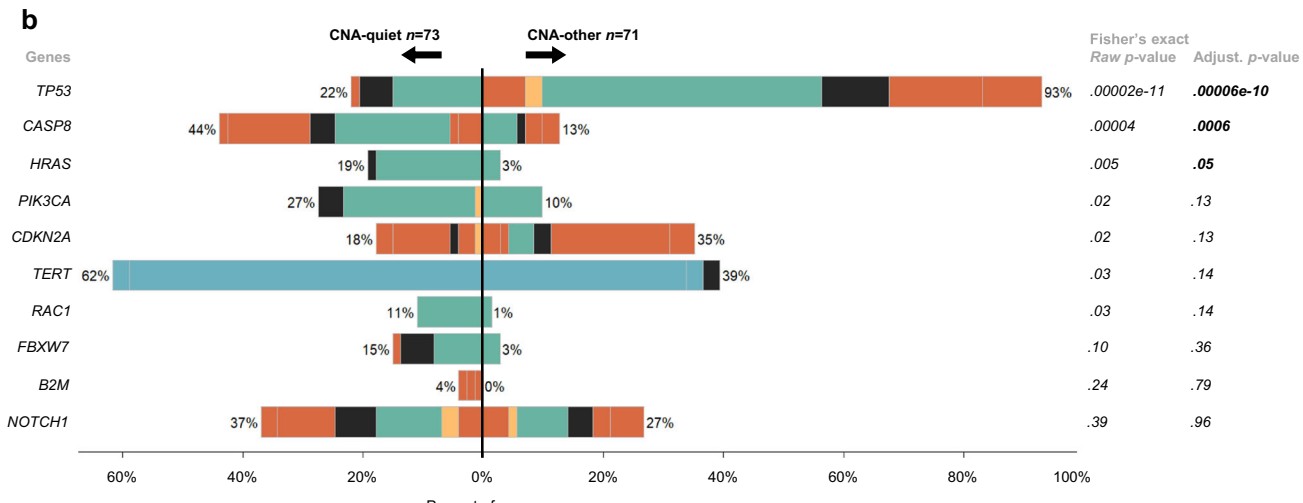

**Fig. 5 | Mutational profile of 73 copy number alteration (CNA)-quiet and 71 CNA-other oral cavity SCC (OCSCC). a** Oncoplot of somatic variants of 15 most prevalent genes in 144 OCSCC samples with 73 CNA-quiet on the left (fraction genome altered (FGA) < 0.20 and < 6 called segments) and 71 CNA-other OCSCC (FGA ≥ 0.20 and/or ≥ 6 called segments) on the right. Colors represent somatic variant classifications, see Supplementary Data 8. **b** Two-sided fisher's exact test to compare mutations between CNA-quiet and -other OCSCC. Raw as well as adjusted p-value (using Benjamini-Hochberg false discovery rate) are displayed; ten genes with lowest p-value are displayed and ordered based on their p-value. Details can be found in Supplementary Data 4. Also, source data are provided with this paper.

enrichment of less frequently occurring mutations in CNA-quiet tumors using TCGA data, which provides mutational data of 15,881 genes (Supplementary Fig. 8, Supplementary Data 7). We did find other mutations in CNA-quiet HNSCC, but after false discovery rate correction none remained other than *TP53, CASP8,* and *HRAS.* Collectively, the major mutational discrepancies detected between the CNA-other and -quiet groups, provide evidence that CNA-quiet tumors form a molecular distinct subclass within HPV-negative HNSCC.

## CNA-quiet oral cavity SCC are clinically distinct with more favorable prognosis

A summary of the patient and tumor characteristics are presented in Table 1, Fig. 6 and Supplementary Fig. 10 (Supplementary Data 2 for a detailed list). The CNA-quiet OCSCC patients were on average older (70 versus 64 years, p < .001). Furthermore, this group contained more females (60% versus 40%, p < .001). Perhaps even more strikingly, the CNA-quiet group included significantly more never smokers (42% versus 19%, p < .001), as well as more never alcohol users (38% versus 20%, p < .001, Figs. 6a–c, Supplementary Fig. 10a). CNA-quiet OCSCC

differed also histologically from CNA-other OCSCC (Fig. 6d–f, Supplementary Figs. 11–12). Specifically, CNA-quiet OCSCC were more often well-differentiated (46% versus 13%), while CNA-other OCSCC were more frequently poorly-differentiated (27% versus 7%, p < .001). Also, while the invasion pattern of CNA-quiet OCSCC was predominantly cohesive (81%), only a minority of CNA-other OCSCC had a cohesive invasion pattern (38%, p < .001). Additionally, we performed a comparative analysis in which pattern of invasion (POI), as reported by Heerema et al.[18], was scored for 26 CNA-quiet and 52 CNA-other OCSCC. POI evidently differed between the groups, with CNA-quiet OCSCC solely being scored as POI-1, -2, or -3, while CNA-other was enriched for POI-4 and -5 (p < .001, Fig. 6d–f). As histological grade is also available in the TCGA, we were able to confirm the enrichment of well-differentiated tumors in the CNA-quiet group (30% well-differentiated in CNA-quiet versus 10% in CNA-other group, p < .001).

The anatomical subsite of CNA-quiet and CNA-other OCSCC within the oral cavity also differed. Specifically, CNA-quiet OCSCC arose more frequently in the vestibule of the mouth and cheek mucosa compared to CNA-other OCSCC (32% and 21% versus 12% and 6%,

**Table 1 | Patient and tumor characteristics of the oral cavity SCC cohort**

| Characteristics | | Patients with OCSCC, No. (%) | | |
| --- | --- | --- | --- | --- |
| | | CNA-other | CNA-quiet | |
| Patients | | 729 (90.9%) | 73 (9.1%) | |
| Age at diagnosis, median (range) | | 64 (21–92) | 74 (15–91) | **0.00003**[a] |
| Sex | Female | 288 (39.5%) | 44 (60.3%) | **0.0009**[b] |
| | Male | 441 (60.5%) | 29 (39.7%) | |
| Tobacco use | Never | 151 (20.7%) | 31 (42.5%) | **0.00002e-1**[b] |
| | Former | 208 (28.5%) | 26 (35.6%) | |
| | Current | 369 (50.6%) | 16 (21.9%) | |
| Alcohol use | Never | 142 (19.5%) | 28 (38.4%) | **0.0004**[b] |
| | Former | 80 (11.0%) | 3 (4.1%) | |
| | Current | 506 (69.5%) | 42 (57.5%) | |
| Anatomical subsite | Cheek mucosa | 43 (5.9%) | 15 (20.5%) | **0.00002e-5**[b] |
| | Floor of mouth | 216 (29.6%) | 6 (8.2%) | |
| | Hard palate | 6 (0.8%) | 2 (2.7%) | |
| | Mobile tongue | 330 (45.3%) | 25 (34.2%) | |
| | Retromolar trigone | 45 (6.2%) | 2 (2.7%) | |
| | Vestibule of mouth | 89 (12.2%) | 23 (31.5%) | |
| Dental status | Dentate | 309 (42.4%) | 27 (37.0%) | 0.27[b] |
| | Edentate | 235 (32.2%) | 28 (38.4%) | |
| | Unknown | 185 (25.4%) | 18 (2.5Z%) | |
| T-stage | T1 | 248 (34.0%) | 30 (41.1%) | 0.48[b] |
| | T2 | 204 (28.0%) | 23 (31.5%) | |
| | T3 | 114 (15.6%) | 7 (9.6%) | |
| | T4a | 149 (20.4%) | 12 (16.4%) | |
| | T4b | 14 (1.9%) | 1 (1.4%) | |
| N-stage | N0 | 451 (61.9%) | 60 (82.2%) | **0.006**[b] |
| | N1 | 93 (12.8%) | 3 (4.1%) | |
| | N2a | 23 (3.2%) | 4 (5.5%) | |
| | N2b | 51 (7.0%) | 3 (4.1%) | |
| | N2c | 25 (3.4%) | 1 (1.4%) | |
| | N3b | 86 (11.7%) | 2 (2.7%) | |
| Disease-stage | I | 209 (28.7%) | 26 (35.6%) | **0.009**[b] |
| | II | 124 (17.0%) | 21 (28.8%) | |
| | III | 121 (16.6%) | 6 (8.2%) | |
| | IVA | 176 (24.1%) | 18 (24.7%) | |
| | IVB | 97 (13.5%) | 2 (2.7%) | |
| | IVC | 2 (0.3%) | 0 (0%) | |
| Tissue invasion | Bone | 77 (10.6%) | 11 (15.1%) | 0.66[b] |
| | Lamina propria | 335 (46.0%) | 36 (49.3%) | |
| | Muscle | 128 (17.6%) | 10 (13.7%) | |
| | Skin | 1 (0.1%) | 0 (0%) | |
| | Submucosa | 17 (2.3%) | 3 (4.1%) | |
| | Unknown | 171 (23.5%) | 13 (17.8%) | |
| Invasion pattern | Cohesive | 254 (34.8%) | 52 (71.2%) | **0.00009e-6**[b] |
| | Non-cohesive | 407 (55.8%) | 12 (16.4%) | |
| | Unknown | 68 (9.3%) | 9 (12.3%) | |
| POI category | POI-1-3 | 28 (3.8%) | 26 (28.0%) | **0.00009**[b] |
| | POI-4-5 | 24 (3.3%) | 0 (0%) | |
| | Unknown | 677 (92.9%) | 47 (64.4%) | |
| Differentiation grade | Well-differentiated | 87 (11.9%) | 31 (42.5%) | **0.00005e-7**[b] |
| | Moderately-differentiated | 403 (55.3%) | 32 (43.8%) | |
| | Poorly-differentiated | 178 (24.4%) | 5 (6.8%) | |
| | Unknown | 61 (8.3%) | 5 (6.8%) | |

**Table 1 (continued) | Patient and tumor characteristics of the oral cavity SCC cohort**

| Characteristics | | Patients with OCSCC, No. (%) | | |
|---|---|---|---|---|
| | | CNA-other | CNA-quiet | |
| Surgical margins | Clear (>5 mm) | 291 (39.9%) | 33 (45.2%) | 0.67[b] |
| | Close (1-5 mm) | 191 (26.2%) | 21 (28.8%) | |
| | Involved (<1 mm) | 133 (18.2%) | 12 (16.4%) | |
| | Unknown | 114 (15.6%) | 7 (1%) | |
| Extranodal extensions (pathological) | Yes | 98 (13.4%) | 6 (8.2%) | |
| | No | 428 (58.7%) | 33 (45.2%) | 0.61[b] |
| | Unknown | 203 (27.8%) | 34 (46.6%) | |
| Treatment type | Radiotherapy (RT) | 18 (5.1%) | 1 (4.2%) | 0.34[b] |
| | Chemoradiotherapy (CRT) | 14 (4.0%) | 0 (0%) | |
| | Surgery | 78 (22.0%) | 9 (37.5%) | |
| | Surgery + RT | 210 (59.3%) | 14 (58.3%) | |
| | Surgery + CRT | 29 (8.2%) | 0 (0%) | |
| | Other | 5 (1.4%) | 0 (0%) | |

OCSCC oral cavity squamous cell carcinoma; T, N and disease stage according to TNM classification 8th edition, 2017. Pathological stage was used when available; when the patient was not treated with surgery, clinical stage was used. Pattern of invasion (POI) as reported by Heerema et al.[18]. For treatment type, only advanced disease-staged (III-IV) tumors with curative intent were included. No patients were treated with cetuximab. [a] p-value obtained by Mann-Whitney test, [b] p-value obtained by Chi-squared test. Unknowns were excluded from the analysis. p < 0.05 indicated in bold.

respectively), while CNA-other OCSCC were more commonly found at the floor of mouth (30% versus 8%, p < .001, Fig. 6h). We compared the dental status within the groups; a comparable number of patients were found to be edentulous (38% in CNA-quiet versus 32% in CNA-other, p = .27). Furthermore, although T-stage did not differ between the groups (p = .48), N- and disease-stage did. Less patients with CNA-quiet OCSCC had N3b staging for regional lymph nodes (3% versus 12%, p = .006, Supplementary Figs. 10b–d). Also, early disease-stage (Stage I–II) was more frequent in CNA-quiet OCSCC (64% versus 46%) while late disease-stage (Stage III–IV) was more common for CNA-other OCSCC (54% versus 36%, p = .002, Fig. 6g).

Surgical margins, tissue invasion, and presence of extranodal extensions did not differ between the groups (Supplementary Figs. 10f–h). Importantly, CNA-quiet OCSCC exhibited markedly better overall survival compared to CNA-other OCSCC (HR 0.44, p = .002, Fig. 6i). Of note, for the survival analysis, patients treated with palliative intent (n = 4 CNA-quiet and n = 49 CNA-other) were excluded, leaving only patients treated with curative intent (n = 69 CNA-quiet and n = 680 CNA-other). In a multivariate model, including all parameters that significantly differed between CNA-quiet and CNA-other, the CNA status remained an independent prognostic variable (Table 2). To disentangle disease-stage, that differed between CNA-quiet and CNA-other groups (Fig. 6g), on survival, we conducted separate survival analyzes for early and late disease-stages (Supplementary Figs. 13a, b). When the analysis was restricted to early-stage lesions, no significant difference was found (p = .22) for CNA-quiet versus CNA-other. However, the more favorable prognosis of CNA-quiet OCSCC was particularly pronounced in late-stage disease (III–IV, p = .02).

**CNA-quiet oral cavity SCC exhibit a less immunosuppressed tumor immune microenvironment**

Multiplex immunohistochemistry (mIHC) was performed to characterize the tumor immune microenvironment (TIME) of 12 CNA-quiet and 27 CNA-other OCSCC (Supplementary Data 9 and 10). Immune cells were found to infiltrate the tumor center (TC), but the highest immune cell densities were found in stromal compartments at the invasive margin (IM), at the border of the tumor, regardless of the CNA class (Fig. 7a-b, Supplementary Fig. 14a). No differences in total immune cell density (T cells, B cells, and CD163+ macrophages together) were found between CNA-quiet and CNA-other OCSCC (Fig. 7b). However, when stratifying per cell type, FoxP3+ regulatory T cells (Tregs) were significantly less frequent in CNA-quiet OCSCC, explained

by a lower fraction in the tumor center (Fig. 7c). Moreover, the ratio of CD8+ cytotoxic T cells, as well as CD4 + T helper cells (CD3+ CD8-FoxP3- T cells), to FoxP3+ Tregs was significantly higher in CNA-quiet OCSCC (Fig. 7d-e), suggesting less immunosuppression by Tregs in the center of these tumors. Correspondingly, the percentage of CD8+ T cells and CD4+ T helper cells within 10 μm radius of a FoxP3+ Treg was lower in CNA-quiet OCSCC (Fig. 7f–h), indicating that less T cells are in the direct neighborhood of FoxP3+ Tregs in CNA-quiet OCSCC. This suggests that T cells in the center of CNA-other OCSCC are more suppressed by the influence of Tregs compared to T cells in CNA-quiet OCSCC. What is more, the percentage of CD8 + T cells as well as CD4+ T helper cells within 10 μm radius of a tumor cell (CD44v6+ ) was higher in CNA-quiet compared to CNA-other OCSCC, especially in the invasive margin (Figs. 7i, j, Supplementary Fig. 14c). In other words, more T cells are closely located to tumor cells, which might suggest increased interaction between T cell and tumor cells, in CNA-quiet OCSCC.

## Discussion

Cancer is caused by an accumulation of somatic genetic and epigenetic alterations. Characteristics are the mutations in specific cancer driver genes, although the genes involved may differ per tumor type. Generally these genetic and epigenetic changes cause a deregulation of signaling networks, as well as metabolic and replication stress. Altogether this causes genomic instability characterized by multiple gains and losses in a variety of chromosomal regions. These CNAs are very typical for most cancers, but remarkably some tumors seem devoid of these genetic changes, a phenomenon observed in a variety of solid cancers[19,20]. In this study, we extensively characterized in a cohort of 802 OCSCC what we called, in line with the TCGA, CNA-quiet oral cancers. We demonstrated that CNA-quiet OCSCC lacking or having few CNAs, should be considered as a separate subclass of HPV-negative HNSCC since they have a different mutational profile (enriched for CASP8 and HRAS mutations with generally wtTP53), other histological characteristics (predominantly cohesive growth pattern and well-differentiated tumors), a less immunosuppressed TIME, and distinct clinical appearance with older patients, more women, more never smokers and alcohol users, and improved overall survival compared to CNA-other OCSCC. We propose to stratify HNSCC in HPV-positive, HPV-negative CNA-quiet, and HPV-negative CNA-other as separate disease entities. Over the past decades, various studies demonstrated that HNSCC can also be classified into groups based on their gene

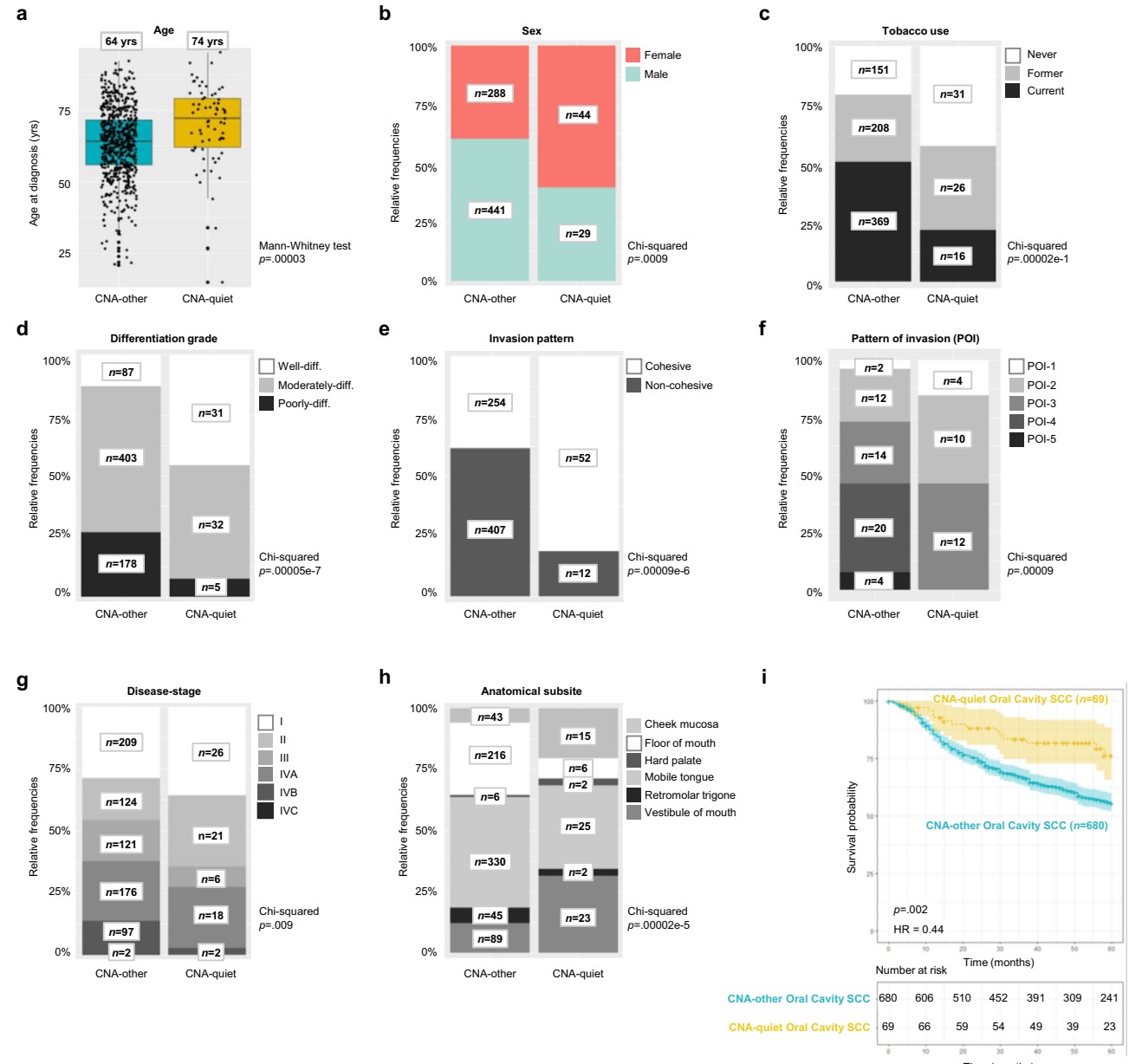

**Fig. 6 | Clinical and histological comparison between copy number alteration (CNA)-quiet (n = 73) and -other (n = 729) oral cavity SCC (OCSCC). a** Age at diagnosis for CNA-quiet and -other OCSCC. *p*-value was obtained using a two-sided Mann-Whitney test. Box shows the interquartile range (IQR) and median (64 yrs for CNA-quiet; 74 yrs for CNA-other). Whiskers extend to 1.5xIQR; outliers are individual points. **b–f**, Relative frequencies and number of patients are depicted in the bar graph per category of (**b**) sex, (**c**) tobacco use, (**d**) differentiation grade, (**e**) invasion pattern, (**f**) pattern of invasion (POI) scored as reported by Heerema et al.[18], in a comparative study the POI of 26 CNA-quiet OCSCC was analyzed and compared with 52 T-stage matched CNA-other OCSCC, Chi-squared test comparing POI1-3 with POI4-5, (**g**) disease-stage, and (**h**) anatomical subsite in CNA-quiet and -other groups. Unknowns were excluded from the analysis. Disease-stage according to TNM classification 8th edition, 2017. Pathological stage was used when available; when the patient was not treated with surgery, clinical stage was used. *p*-values were obtained using a Chi-squared test. **i** Kaplan-Meier curve for CNA-quiet (n = 69) and -other OCSCC (n = 680) with *p*-value and hazard ratio (HR) displayed, obtained by log-rank test. Patients with curative treatment intent were included in the survival analysis; patients with palliative intent (n = 4 CNA-quiet and n = 49 CNA-other) were excluded. Source data are provided with this paper.

expression profiles[8,21–23]. Although these studies are highly relevant as they demonstrate molecularly deviating groups, a consensus classification has not been reached. We propose a two-level classification within HPV-negative HNSCC: first at the genetic level and second on expression profiles. It has been shown that within HPV-positive tumors, it is clinically and biologically informative to stratify the genetically-defined subgroups by expression profiling[24–26].

An OCSCC cohort more than twice the size of the TCGA cohort allowed us to find additional cancer driver genes that are frequently affected in CNA-quiet OCSCC, notably *PIK3CA, RAC1,* and *TERT*, besides the previously established HRAS and CASP8. Some of those CNA-quiet enriched mutations might underlie the low level of genomic instability. The association of *CASP8* mutations and an CNA-quiet genotype has been observed in other studies[27], but a causal relationship is not easily explained. Conversely, mutations in *TP53* permit genomic instability, leading to whole genome doubling events and excessive CNAs, which might explain that wt*TP53* is a prerequisite for a CNA-quiet genotype[28–30]. The addition of analyzes on *TERT* promoter mutations (which are lacking in most of TCGA data) in our study highlighted their particularly high prevalence in OCSCC in general[16,31], and in CNA-quiet

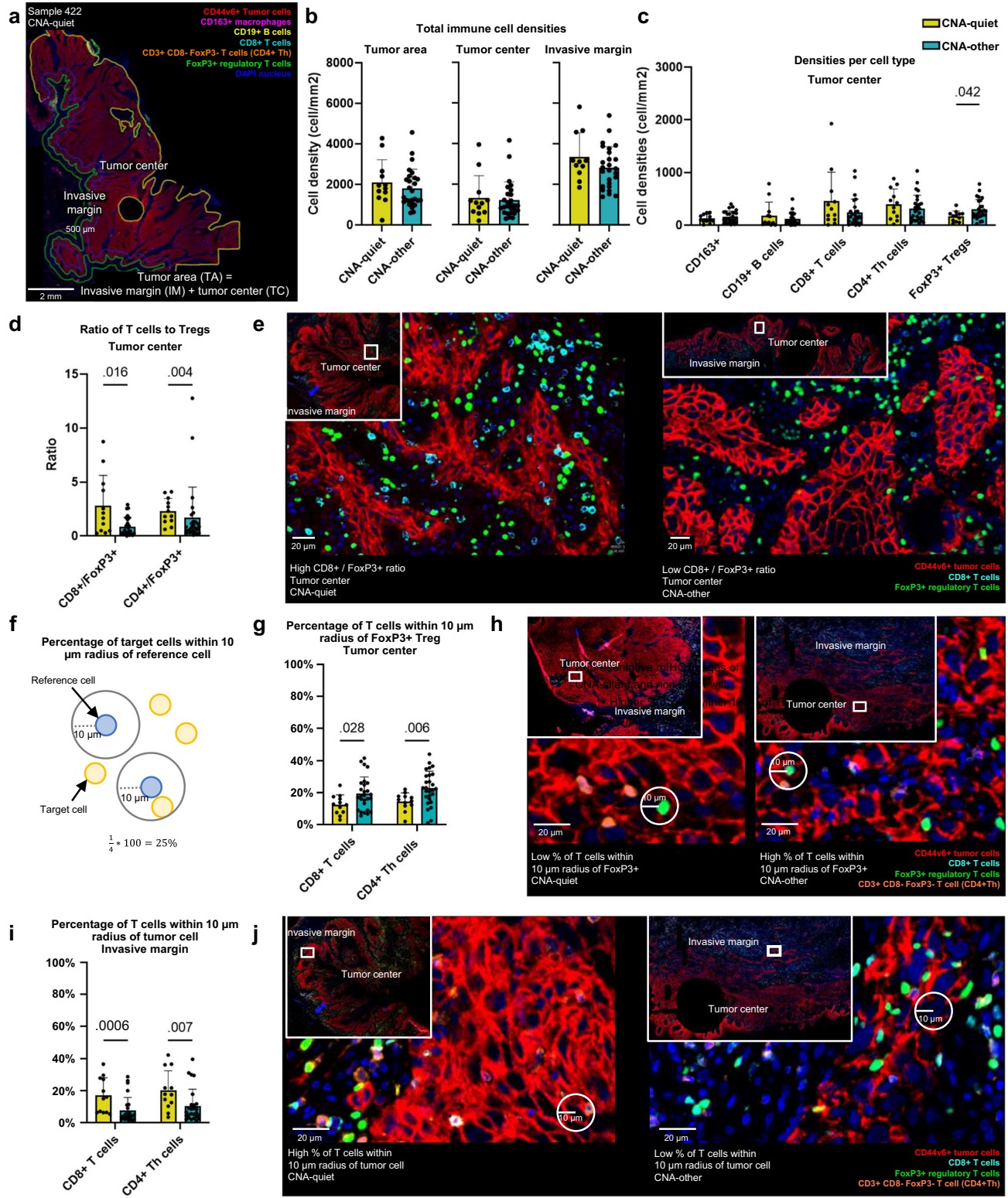

OCSCC specifically. The stabilization of telomeres is a crucial step in carcinogenesis.

We confirmed that *CASP8* mutations are highly enriched in the CNA-quiet group. We reviewed the nine cases in the CNA-other group with *CASP8* mutations. Mutated *CASP8* was also significantly associated with low FGA in the CNA-other group, and one of these *CASP8*-mutated CNA-other cases had even an FGA of less than 0.20, but had just a few too many segments. This suggests that besides FGA and number of segments, also the *CASP8* mutation status might be

considered as an important determinant of the CNA-quiet group, and that the CNA-quiet group should be defined as either *CASP8* mutated or FGA < 0.20 and number of segments below six when applying lcWGS.

In this study we demonstrated that CNA-quiet OCSCC had a better overall survival compared to CNA-other OCSCC. To comprehend this clinical benefit, we characterized histological parameters as well as the TIME. First of all, CNA-quiet OCSCC evidently differed from CNA-other OCSCC in their histological appearance as they generally had a

**Fig. 7 | Tumor immune microenvironment of copy number alteration (CNA)-quiet (n = 12) and CNA-other (n = 27) oral cavity SCC (OCSCC). a** Representative image stained with a seven-color multiplex immunohistochemistry (mIHC) Opal panel characterizing CD44v6+ tumor cells, CD163+ macrophages, CD19+ B cells, CD8+ T cells, CD3+ CD8- FoxP3- T cells (CD4+ T helper cells), and FoxP3+ regulatory T cells (Tregs, CD3 + , CD8-). Immune cell densities were determined within the tumor center (TC), invasive margin (IM), consisting of a 250 µm inner and outer margin, as well as the tumor area (TA), which is the sum of the TC and IM. **b** Total immune cell (T cells, B cells, and CD163+ macrophages together) densities (in cells/mm2) in tumor area, tumor center, and invasive margin of CNA-quiet (n = 12, yellow) and CNA-other (n = 27, blue) OCSCC. **c** Densities (in cells/mm2) per immune cell type in the tumor area of CNA-quiet (n = 12) and CNA-other OCSCC (n = 27). **d** Ratio of the densities CD8+ T cells and CD4+ Th cells to FoxP3+ Tregs in the tumor center of CNA-quiet (n = 12) and CNA-other OCSCC (n = 27).

**e** Representative mIHC image of CNA-quiet (left panel) and CNA-other tumor (right panel) with respectively high and low CD8+ T cell to FoxP3+ T cell ratios. **f** Percentage of target cells (yellow) within 10 µm radius of a reference cell (blue). **g** Percentage of CD8+ T cells and CD4+ Th cells within 10 µm radius of a FoxP3+ Treg in the center of CNA-quiet (n = 12) and -other OCSCC (n = 27). **h** Representative mIHC image of CNA-quiet (left panel) and CNA-other tumor center (right panel) with respectively low and high percentage T cells within 10 µm radius of a FoxP3+ Treg. **i** Percentage of CD8+ T cells and CD4+ Th cells within 10 µm radius of a tumor cell in the invasive margin of CNA-quiet (n = 12) and CNA-other OCSCC (n = 27). **j** Representative mIHC image of CNA-quiet (left panel) and CNA-other tumor center (right panel) with respectively high and low percentage T cells within 10 µm radius of a tumor cell. p-values were obtained by multiple unpaired non-parametric two-sided Mann-Whitney tests. Data are presented as mean and error bars represent standard deviation. Source data are provided with this paper.

cohesive growth pattern and are more often well-differentiated versus a non-cohesive invasion pattern and a moderate to poor differentiation grade, frequently observed in CNA-other OCSCC. A cohesive growth pattern has been shown to be associated with less locoregional recurrences and improved disease-free survival for HNSCC[32]. Secondly, early disease-stage was more abundant for CNA-quiet OCSCC, evidently also contributing to the difference in prognosis as indicated by the multivariate model. Interestingly, the difference in prognosis between CNA-quiet and CNA-other OCSCC was notably accentuated during late-stage disease. Remarkably, none of the advanced disease-stage CNA-quiet OCSCC received chemoradiotherapy (CRT) or surgery with postoperative CRT (Supplementary Fig. 10e), presumably due to their more favorable clinical and histological characteristics.

HPV-negative CNA-quiet OCSCC and HPV-positive HNSCC have some shared clinical characteristics (improved prognosis, generally non-smoking). However, HPV-positive HNSCC mostly occurs in the tonsillar crypt epithelium in the oropharynx. While we acknowledge some similarities, HPV-positive OPSCC is considered a separate disease entity when compared to HPV-negative OPSCC, both at the clinical and biological level. Within OCSCC, HPV rarely occurs and does not play a prognostic role, as reported by Nauta et al.[33]. We analyzed the survival of HPV-positive OCSCC, CNA-other, and CNA-quiet OCSCC and only CNA-quiet OCSCC showed a more favorable outcome as expected (Supplementary Fig. 13c). Taken together, while for OPSCC the difference between HPV-negative and HPV-positive is critical, for OCSCC, the distinction between CNA-quiet and CNA-other is crucial. Whether treatment de-escalation strategies, such as reduced radiotherapy schedules as applied for HPV-positive OPSCC[34], might become applicable to CNA-quiet tumors, remains to be determined.

Using mIHC, we showed that CNA-quiet OCSCC had a more favorable TIME. Based on previous studies on correlations between genomic instability and immune signatures in HNSCC[19,35–38] and other solid tumors[39–41], we hypothesized to find a more immune-infiltrated TIME for CNA-quiet OCSCC. Specifically, it has been reported that many CNAs in the genome negatively correlated with interferon-γ signaling[19,35,36], the expression of T cell markers[19,36], B cell and NK cell markers[19], as well as immune cell infiltration[36,37]. Of note, those studies were merely based on deconvolution analyzes from gene expression data[19,35–38], and generally not on spatial analysis of the TIME. We found, using mIHC, that CNA-other OCSCC exhibited a more immunosuppressed TIME compared to CNA-quiet OCSCC. These observed differences in the TIME of CNA-quiet and CNA-other OCSCC might contribute to the clinical benefit seen in OCSCC with low or few CNAs. The distinct mutational profiles of the CNA-quiet and CNA-other groups could underlie their immunological differences. Specifically, TP53 mutations have been correlated with an immunosuppressed microenvironment[42–45]. Conversely, it has been shown that CASP8 mutations appear to be enriched in CD8+ T cell inflamed HNSCC[46,47], further indicating that mutational differences between the CNA groups might contribute to their deviating TIME.

CNA-quiet OCSCC typically occurs in non-smoking older women. Given the older age at which CNA-quiet OCSCC are diagnosed (median 74 years compared to 64 years for CNA-other), this subclass—now found to constitute 9.1% of the OCSCC—may well increase within aging populations, although it should be noted that this proportion is based on two cohorts in the Netherlands. It is unclear whether another etiological factor is in play. HPV does not play a role in oral cancer and can be excluded[33]. As we only had access to FFPE samples and did not perform whole exome sequencing (WES), we could not analyze mutational signatures that might point to a certain etiological factor, reported by Alexandrov et al.[48]. Specifically, these signatures are based on mutational patterns likely associated with biological processes or etiological factors, with, for example, C > A transversions being linked to smoking and C > T transitions to ultraviolet light (UV) and age[48]. We hypothesize that CNA-quiet OCSCC might fall more into the mutational signature associated with age whereas CNA-others might be more linked to the smoking signature.

The site of origin within the oral cavity differed between the groups, with CNA-quiet OCSCC more frequently occurring at the vestibule of the mouth and cheek mucosa. This raises a question about the etiology of CNA-quiet OCSCC. However, a comparable fraction of patients were edentulous, excluding a potential effect of dentures. Furthermore, the location of the tumor might possibly explain the observed differences in N-stage between the groups as certain tumor sites might have greater tendency to metastasize to lymph nodes. Lastly, the tumor site might also impact surgical options and how radical the tumor can be excised. However, no differences were found between the groups with respect to surgical margins, tissue invasion or presence of extranodal extensions.

Immune checkpoint inhibitors (ICI) targeting PD-1 is a treatment option for recurrent and metastatic HNSCC, however, the response rate remains below 20%[49–51]. Also in trials with neoadjuvant ICIs, response rate are typically 20−35%[52]. Accordingly, there is a need to stratify patients who will likely respond from those who will not. Among the OCSCC, we hypothesize that CNA-quiet cancers might be the group benefitting most from ICIs. For melanoma, it has been shown that patients with low copy number burden benefit more from either single agent anti-CTLA-4 or anti-PD-1, or anti-CTLA-4 followed by anti-PD-1[19,41,53–55]. Likewise, having few CNAs was linked to a durable clinical response to anti-PD-1/PD-L1 for patients with non-small cell lung cancer[40,56], lung adenocarcinoma[57], and gastrointestinal cancer[39]. While this link between CNAs and ICI response seems evident for those other solid tumors, whether CNAs are associated with ICI response in HNSCC remains to be elucidated. We hypothesize that CNA-quiet OCSCC might benefit from immunotherapy and we would strongly encourage to perform genomic profiling to stratify for CNAs in past and future trials.

In conclusion, we demonstrated in a large independent cohort that HPV-negative HNSCC can be classified based on copy number data and mutational profiles. These subclasses show clear discrepancies in

**Table 2 | Multivariate analysis of the multicenter oral cavity SCC cohort**

| Final classification | | Hazard ratio | p-value |
|---|---|---|---|
| | CNA-other | Reference | Reference |
| | CNA-quiet | 0.55 | **0.04** |
| Age | | 1.03 | **0.00001e-3** |
| Sex | | | |
| | Female | Reference | Reference |
| | Male | 1.25 | 0.10 |
| Tobacco use | | | |
| | Never | 0.64 | **0.02** |
| | Former | 0.91 | 0.49 |
| | Current | Reference | Reference |
| Alcohol use | | | |
| | Never | 1.20 | 0.32 |
| | Former | 1.28 | 0.20 |
| | Current | Reference | Reference |
| Anatomical subsite | | | |
| | Cheek mucosa | Reference | Reference |
| | Floor of mouth | 0.78 | 0.38 |
| | Hard palate | 0.93 | 0.90 |
| | Mobile tongue | 0.85 | 0.53 |
| | Retromolar trigone | 0.74 | 0.37 |
| | Vestibule of mouth | 0.97 | 0.93 |
| N-stage | | | |
| | N0 | Reference | Reference |
| | N1 | 1.07 | 0.76 |
| | N2a | 1.49 | 0.25 |
| | N2b | 1.51 | 0.15 |
| | N2c | 2.01 | **0.05** |
| | N3b | 2.56 | 0.08 |
| Disease-stage | | | |
| | I | Reference | Reference |
| | II | 1.40 | 0.12 |
| | III | 2.22 | **0.001** |
| | IVA | 2.25 | **0.002** |
| | IVB | 2.18 | 0.15 |
| Invasion pattern | | | |
| | Cohesive | Reference | Reference |
| | Non-cohesive | 1.22 | 0.17 |
| | Unknown | 1.53 | 0.12 |
| Differentiation grade | | | |
| | Well-differentiated | 0.72 | 0.14 |
| | Moderately-differentiated | Reference | Reference |
| | Poorly-differentiated | 1.11 | 0.44 |
| | Unknown | 1.03 | 0.91 |

P-values obtained by logrank test on Cox proportional hazard model. Only patients with curative treatment intent (n = 69 CNA-quiet and n = 680 CNA-other) were included for the analysis; patients with palliative intent (n = 4 CNA-quiet and n = 49 CNA-other) were excluded. p < 0.05 indicated in bold.

histological and clinical parameters, tumor immune microenvironment, and prognosis. Whether these two groups respond differently to therapeutic regimens such as immunotherapy remains to be determined, but we encourage that genetic profiling, including copy number analysis, should be standard in clinical trial evaluation and design.

## Methods

### Ethical approval and consent to participate
The study protocol for this retrospective study was approved by the Institutional Review Board (IRB) at Amsterdam UMC location VUmc under number 2021-0511. Signed informed consent was not required according to the IRB because of the retrospective nature of the study, the number of patients no longer alive, the notion that none of the patients objected to the secondary use of available clinical data and tissue samples for research, and pseudo-anonymization of all presented data. Privacy guidelines were followed according to the EU General Data Protection Regulation.

### Acquisition and filtering of the cancer genome atlas dataset
Publicly available clinical data of 530 HNSCC samples from the cancer genome atlas cohort were downloaded from cBioPortal[9,58]. First, samples from metastases were removed, leaving only primary tumor samples. Next, we categorized the reported primary tumor site into four major anatomical sites: oral cavity, oropharynx, hypopharynx, and larynx. Specifically, alveolar ridge, buccal mucosa, floor of mouth, hard palate, and lip were grouped as oral cavity; base of tongue and tonsil as oropharynx. Furthermore, since this study focused solely on HPV-negative tumors, we filtered out HPV-positive tumors based on detection of viral transcripts, as reported by Nulton et al.[59]. Also, the p16 immunohistochemistry (IHC) status in the clinical data from TCGA was checked for the remaining non-HPV-positive oropharyngeal tumors and also these were removed if they appeared to be p16-positive or NA.

Segment data derived from Affymetrix SNP 6.0 array were downloaded from cBioPortal[9]. Furthermore, MAF files were acquired via the genomic data commons (GDC) portal[60] using GDCquery_Maf of the TCGAbiolinks R package. Purity and ploidy estimates of samples using ABSOLUTE were obtained from supplemental data from the 2018 Pan-Cancer Atlas publications, available through the GDC website[60].

### Recalculation of the fraction genome altered by correcting for stromal contamination
In order to correct for stromal contamination, a consensus fit for tumor purity and ploidy was derived using R packages ACE[10] and ABSOLUTE[11]. This was defined as the ACE fit with the lowest corrected error score, with the correction factor depending on the distance from the ABSOLUTE fit as calculated by the formula $\sqrt{(\text{purity}_{ACE} - \text{purity}_{ABSOLUTE})^2 + (\text{ploidy}_{ACE} - \text{ploidy}_{ABSOLUTE})^2/5}$. The 5 in this formula reduces the weight of ploidy difference so its contribution to the correction is in balance with purity difference. Segment values were recalculated based on the new purity and ploidy estimates. A segment was considered altered when its copy number value, adjusted for ploidy and purity, differs more than 0.5 from the most common copy number in the tumor. FGA was calculated as the total genomic length of altered segments divided by the total genomic length of all segments.

### Patients and specimens of oral cavity squamous cell carcinoma cohort
We obtained 900 FFPE tumor specimens from a consecutive cohort of TNM-8 stage I-IV HPV-negative OCSCC patients diagnosed between 2008 and 2014 at Amsterdam UMC location VUmc and Erasmus MC Rotterdam. Sex was used in the current study, as reported in the patient file. Gender has not been documented or used in the study. Both female and male sex were included. A core biopsy of 1 mm diameter was taken from the tumor area of FFPE specimen by guidance of the corresponding hematoxylin & eosin (H&E)-slide using a plunger system (Kai Europe GmbH, Solingen, Germany). Genomic DNA from the core biopsies was isolated using the NucliSENS easyMag

(bioMérieux, Marcy-l'Étoile, France). DNA yield was measured using Qubit broad range (BR) kit (Thermo FisherScientific).

## Multiplex Ligation-dependent Probe Amplification

Using the TCGA HNSCC dataset and own high resolution microarray comparative genomic hybridization (CGH) datasets, we selected and validated ten DNA regions most frequently gained or lost in HPV-negative HNSCC: 2q36 (*CUL3*), 3q26 (*PIK3CA* and *SOX2*), 4q35 (*FAT1*), 5q15 (*KIAAO825*), 7p11 (*EGFR*), 8q11 (*SNAI2*), 9p22 (*NFIB*), 11q13 (*CCND1* and *FADD*), 18p11 (*TGIF1*), and 20q11 (*BCL2L1*). Based on these regions, the SALSA MLPA probe mix P477 head and neck carcinoma version A1 was developed by MRC Holland with 36 classification probes. Additionally, this probe mix also contained 13 reference probes for data normalization based on chromosomal regions least frequently changed by copy number in HPV-negative HNSCC (Supplementary Data 11).

DNA from FFPE tumor core biopsies was used for MLPA with probemix and SALSA MLPA EK1 reagent kit (EK1-FAM, MRC Holland), following the manufacturer's instructions (MLPA general protocol MDP v007). In brief, up to 50 ng DNA per sample was diluted in 5 μL low Tris EDTA (TE) buffer (10 mM Tris-HCl, pH 8.0 + 0.1 mM EDTA) followed by an incubation of 5 minutes at 98 °C in a thermal cycler for DNA denaturation. For every experiment, low TE buffer was taken along as negative control and three controls from healthy individuals were used as normal reference. The probe mix was added to the sample, incubated for 1 minute at 95 °C and hybridized for 16 to 20 hours at 60 °C. Next, probe ligation was performed by a 15 minute incubation at 54 °C followed by 5 minutes at 98 °C (for ligase inactivation). The SALSA PCR primer mix and polymerase (provided in the kit) were then added and PCR amplification was executed by 35 cycles of 30 seconds at 95 °C, 30 seconds at 60 °C, 60 seconds at 72 °C, followed by a final step of 20 minutes at 72 °C and till further processing at 15 °C. Fragment analysis by capillary electrophoresis was performed using the ABI 3500 Genetic Analyzer (Applied Biosystems) using Hi-Di formamide (Applied Biosystems) and LIZ GS 600 size standard (Applied Biosystems).

For data analysis, Coffalyser.Net software (v.140721.1958, MRC Holland) was used. We scored a region as changed in case of two aberrant probes in the same direction (outside the normal range 0.7–1.3). Since the regions 3q and 11q were represented by more probes, three probes had to be increased or decreased to be scored as aberrant. Furthermore, if there was a decreased probe in an increased region or vice versa, another probe had to be decreased for the region to be counted as decreased. When more than two regions were changed, the tumor was classified as definitive CNA-other. In the case of two or less regions changed, the tumor was categorized as CNA-quiet candidate and lcWGS followed to determine final CNA classification.

## Low-coverage whole-genome sequencing library preparation

Up to 100 ng DNA from FFPE tumor core biopsies was used for lcWGS library preparation. Libraries were generated using the KAPA Hyper-Plus Kit (catalog No. 07962428001 KAPA biosystems), following the manufacturer's instructions (KAPA HyperCap Workflow v3.0). In brief, enzymatic fragmentation was executed by an incubation of 15 minutes at 37 °C in a thermal cycler, followed by end repair and A-tailing for 30 minutes at 65 °C, and index adapter ligation for 15 minutes at 20 °C using 0.03 μM index adapter from integrated DNA technologies (IDT). The adapters contained 10 nt unique dual index (UDI) barcodes for sample identification and unique molecular identifiers (UMIs) for template identification. After ligation, a 0.8X AMPure XP bead (Beckman Colter) cleanup was performed. Subsequently, universal KAPA Library Amp Primer Mix was added and incubated in a thermal cycler with the following program: 45 seconds at 98 °C, 8–9 cycles of 15 seconds at 98 °C, 30 seconds at 60 °C, 30 seconds at 72 °C, followed by a final step of 1 minute at 72 °C and further processing at 4 °C. Next,

amplified DNA was purified by two rounds of 1X AMPure XP bead cleanup with elution in 10 mM Tris-HCL, pH 8.0. DNA yield of libraries was measured using a Qubit high sensitivity (HS) kit and a quality control was performed using an Agilent Bioanalyzer HS chip (Agilent Technologies). Samples were equimolarly pooled and sequenced using the S4.300 kit with paired-end 150 bp on an Illumina NovaSeq 6000 with approximately 10 M reads per sample.

## Low-coverage whole-genome sequencing analysis

Sequence reads were mapped to reference genome hg19 using Burrow-Wheeler Aligner (BWA) mem version 0.7.17[61]. Mapped reads were binned (bin size 500 kbp) using HMMcopy version 1.42[62] followed by normalization using QDNAseq version 1.36[63]. Background patterns were inferred from data of 42 FFPE-derived normal controls[14] using principal component analysis, and background contribution of the two main principal components was subtracted from each sample. Segmentation was done with DNAcopy version 3.17[63] and copy number plots as well as purity and ploidy fits were made using ACE version 1.18[10]. Segment calling was performed as described before, with 0.02 as abscutoff[14]. The FGA was calculated as the summed length of called segments divided by the summed length of all segments. Sex chromosomes were excluded from analysis. OCSCC were classified as CNA-other if the FGA was ≥ 0.20 and/or ≥ 6 segments were called.

## Target-enrichment sequencing library preparation

The DNA sample libraries obtained from the KAPA HyperPlus kit sample workup were used for target-enrichment sequencing using the KAPA HyperCapture Reagent kit (9075810001) and KAPA HyperCapture Bead kit (9075780001), following the manufacturer's instructions (KAPA HyperCap Workflow v3.2). In brief, 24 samples per capture library were mixed together with a similar mass ( ~ 62.5 ng of each sample). A panel capturing the following 29 genes was used: *AJUBA*, *ASXL1*, *B2M*, *CASP8*, *CDKN2A*, *DDX3X*, *EPHA2*, *FAT1*, *FBXW7*, *FGFR3*, *HRAS*, *KDM6A*, *KEAP1*, *KMT2D*, *KRAS*, *NFE2L2*, *NOTCH1*, *NOTCH2*, *NSD1*, *PIK3CA*, *PTEN*, *RAC1*, *RB1*, *RHOA*, *TERT* (promoter only)*, TGFBR2*, *TP53*, *TP63*, *ZNF740*. Hybridization was performed by incubation in a thermal cycler for 5 minutes at 95 °C, followed by 16 to 20 hours at 55 °C. The hybridized DNA was captured using KAPA HyperCap Capture Beads (cat. no. 09075780001, Roche, Basel, Switzerland) and the DNA-beads mixture was washed to remove unbound DNA. Bound DNA was eluted and amplified by PCR using adapter-specific primers in a thermal cycler with the following program: 45 seconds at 98 °C, 16 cycles of 15 seconds at 98 °C, 30 seconds at 60 °C, 30 seconds at 72 °C and a final step of 1 minute at 72 °C and final at 4 °C until further processing. Next, a 1X AMPure XP bead cleanup with elution in PCR grade water was executed. The amplified enriched DNA library yield was measured using Qubit HS kit and a quality control performed using an Agilent Bioanalyzer HS chip. The DNA capture pool was sequenced using the S4.300 kit with paired-end 150 bp on an Illumina NovaSeq 6000 with approximately 14 M reads per sample.

## Target-enrichment sequencing analysis

FASTQ files were turned into unmapped BAM files using fgbio (v2.0.3) FastqToBam to extract UMIs from read names. The unmapped BAM files were mapped to the hg19 reference genome using BWA mem version 0.7.17[61] and UMI info was added using fgbio ZipperBams. Next, reads were grouped and marked using fgbio GroupReadsByUmi and CallMolecularConsensusReads. The resulting BAM file with consensus reads was mapped again and reads with a minimum family size of three reads were extracted using fgbio FilterConsensusReads. Finally, overlap in paired reads was removed using fgbio ClipBam. Somatic variants were called using VarScan v2.4.4 (min-reads2 3, min-var-freq 0.02, *p*-value 0.9)[64] and GATK v4.3.0.0 Mutect2 (min-allele-fraction 0.02, min-reads-per-strand 2)[65] and annotated using GATK Funcotator.

Variants had to be called by both callers. An additional filtering step that tests the likelihood that variants are caused by locus- and sample-specific errors was performed as described before[66]. Variants were considered germline SNPs when gnomAD_genome_AF > = 0.01 or dbSNP_COMMON = 1 or present in > 5 cases in a panel of normals (PON, n = 1762) from Hartwig Medical Foundation. Finally, intronic, silent and UTR variants were removed, with exception of *TERT* promoter mutations and *TP53* 5' UTR mutations.

Using the list of filtered variants and the sample purity and ploidy estimate, absolute copies of the variants were calculated and plotted on the copy number profile (Supplementary Fig. 6). Purity of samples without CNAs was determined on the basis of variant allele frequency of somatic variants. Samples with neither somatic variants nor CNAs were excluded from further analyzes. In order to review that all germline SNPs were correctly filtered out, the variants were plotted on the copy number profile as being a somatic variant or a germline variant. Inferred number of somatic and germline copies was calculated using the linkvariants function of ACE. If a variant fitted best as germline, the Catalog Of Somatic Mutations In Cancer (COSMIC)[60] was consulted and the variant was filtered out as germline SNP unless the variant was previously reported as somatic variant confirmed. In that case it was considered as somatic variant. Only somatic variants that remained after filtering are reported. Oncoplots, oncobars and mutually exclusive heatmaps were made using package Maftools version 2.14.0.

### Power analysis for CNA and variant calling

With the final model for tumor purity, the power for calling segments was calculated per segment using the inferred distribution of normal control samples on that segment, the standard deviation of the segment value of the sample, and the expected segment value observed with a single gain or loss given the purity. Samples were required to have > 80% power over at least 80% of the genome. Power to detect somatic variants was derived per genomic location represented in the capture panel by calculating the expected VAF of a single copy variant present in all tumor cells based on the total number of copies of the respective locus and the tumor purity. Power was determined as the probability to call the variant given the expected VAF, the local sequencing depth, and the expected sequencing error rate (see above) using the binomial distribution.

### CNA intratumor heterogeneity in multiple biopsies

Multiple tumor core biopsies were punched in different blocks for nine CNA-quiet OCSCC, according to aforementioned methods used for the entire cohort. After measuring DNA yield, lcWGS and targeted sequencing was performed according KAPA HyperCap FFPE DNA Workflow v1.1 with nine PCR cycles to obtain copy number plots and somatic mutations present in the tumor core biopsies to test for CNA intratumor heterogeneity.

### Multiplex immunohistochemistry Opal staining

FFPE tissue was cut in 4 μm thick sections on superfrost plus adhesion microscope slides (Epredia, J1800AMNZ) for manual mIHC staining. First, deparaffinization was performed according to standard protocol. In brief, after 60 minutes incubation at 60 °C, the sections were sequentially immersed in xylene (two times 7.5 minutes), 100% ethanol (5 minutes), and 97% ethanol (5 minutes). To block endogenous peroxidase, the slides were exposed to a mixture of methanol and 0.3% hydrogen peroxidase ($H_2O_2$) for 20 minutes. Subsequently, they were washed in 70% ethanol for 5 minutes, followed by a rinse in MilliQ. The tissue sections were fixed by immersing the slides in 10% neutral buffered formalin for two hours in order to prevent the tissue from detaching from the microscopic glass slide. After fixation, the sections were washed with agitation (90 rpm) in MilliQ water and in Tris-buffered saline with 0.05% Tween−20 (Sigma-Aldrich, P2287) (TBST).

For heat-mediated antigen retrieval, the sections were transferred into an antigen retrieval buffer consisting of Tris-EDTA (10 mM Tris, 1 mM EDTA, pH9) and ProClin300 (Sigma-Aldrich, 48912-U). The sections were heated in the microwave on 100% power until boiling, with a duration of 4.2 minutes, followed by 15 minutes on 30% power. To cool down, the slides were placed in ice water for 15 minutes and washed in MilliQ with agitation (90 rpm). A hydrophobic pen (Vector Laboratories, ImmEdge, H-4000) was applied around the tissue sections, followed by a wash in TBST with agitation (90 rpm).

The protocol for primary and secondary antibody (Ab) incubation consisted of a repetitive cycle, with sequence information and Ab details outlined in Supplementary Data 9. Each incubation step was conducted in a dark, humidified chamber at room temperature or 4 °C for the overnight Ab incubation. At the beginning of each cycle, the tissue sections were blocked using antibody diluent/block (Akoya, ARD1001EA) for 10 minutes. Next, sections were incubated with a primary Ab and antibody diluentblock and washed in TBST (three times 2 minutes) with agitation (90 rpm). For primary Ab CD19 (cycle 2) blocking consisted of a 30 minute incubation in a mixture of 20% normal goat serum (NGS) (Genetex, GTX3206) and antibody diluent/block. Next, the slides were incubated with a ready to use horseradish peroxidase (HRP) conjugated secondary Ab (Akoya, ARH1001EA). To enhance the reaction for CD19 (cycle 2), the slides were first incubated with a biotin-labeled secondary Ab and later with a HRP-conjugated Streptavidin (Dako, P0397). After secondary Ab incubation the slides were washed in TBST (three times 2 minutes) with agitation (90 rpm). Subsequently, a fluorophore labeled tyramide (TSA) diluted in amplification diluent (Akoya, FP1498) was added interacting with the HRP-conjugated secondary Ab and generating the Opal signal amplification. After 10 minutes of incubation the slides were washed in TBST (three times 2 minutes) with agitation (TBST).

Ab stripping was performed to remove the primary Ab and secondary Ab and any non-specific staining after each cycle. The sections were heated in TRIS-EDTA antigen retrieval buffer as described in Antigen retrieval. An exception was made after cycle 5, when denaturation solution kit (Biocare medical, DNS001L) was used for Ab stripping (Supplementary Data 9). Cell nucleus staining was performed by incubating with DAPI (4 drops per ml TBST, Akoya) for 5 minutes, after which slides were washed in TBST and subsequently MilliQ water. The sections were mounted in prolong diamond antifade mount (Invitrogen, P36970) and covered with microscope cover glasses (VWR, ECN631-1575).

### Multiplex immunohistochemistry visualization and data analysis

The sections were scanned using the Vectra Polaris (Akoya). mIHC staining and scanning was performed in batches of approximately ten slides. An optimal customized scanning protocol was developed for each batch of slides which was created by calculating the median exposure time for all Opal markers, DAPI and autofluorescence at various spots on one slide in each staining batch. Inform 2.6 was utilized for the extraction of unmixed signals, and background staining was eliminated by using unstained negative controls. Qupath version 0.4.3[67] was used for stitching, tissue annotation, cell detection, cell phenotyping, pixel classification, and spatial analysis. After manually outlining tumor fields, the invasive margin was automatically annotated with 250 μm radius (Fig. 7a). A trained head and neck pathologist reviewed the sections for tumor presence. A pixel classifier was trained to further segment the tissues into tumor field and stroma within the abovementioned annotations. Tissue sections with low-quality staining were excluded.

### Statistical analysis

Statistical analyzes were performed using GraphPad Prism 9.3.1 software or R version 4.2.3. Two-tailed paired nonparametric Wilcoxon

rank-sum test and unpaired nonparametric Mann-Whitney tests were used to compare two paired and unpaired groups, respectively. To test for normality Shapiro-Wilk normality test was used. Chi-squared test was used for comparisons of multiple categories. Survival analysis was executed using the package survival version 3.5-5 and visualized by the package survminer version 0.4.9.

## Reporting summary
Further information on research design is available in the Nature Portfolio Reporting Summary linked to this article.

## Data availability
The raw sequencing data generated in this study have been deposited in the European Genome-phenome Archive (EGA) under accession number EGAD50000000790 via [https://ega-archive.org/datasets/EGAD50000000790]. Data will be made available under a data transfer agreement that will only contain statements on acknowledgment to the source publication in manuscripts using the data, commercial use of the data (not allowed), transfer of the data to other parties (not allowed), the aim of the study and the estimated time required for the planned analyzes, in practice 1 to 3 years, but as long as needed. The mIHC imaging data generated in this study is deposited in the bio-image archive database and available under the accession number S-BIAD1352 via [https://www.ebi.ac.uk/biostudies/BioImages/studies/S-BIAD1352]. Publicly available clinical data of 530 HNSCC samples from TCGA were downloaded from cBioPortal and available via [https://www.cbioportal.org/study/clinicalData?id=hnsc_tcga]. Segment data derived from Affymetrix SNP 6.0 array were downloaded from cBioPortal available via [https://www.cbioportal.org/study/summary?id=hnsc_tcga]. MAF files were acquired via the genomic data commons (GDC) portal using GDCquery_Maf of the TCGAbiolinks R package. Purity and ploidy estimates of samples using ABSOLUTE were obtained from supplemental data from the 2018 Pan-Cancer Atlas publications, available through the GDC website via [https://portal.gdc.cancer.gov/projects/TCGA-HNSC]. Source data are provided with this paper.

## Code availability
Binned read counts, targeted sequencing depth, filtered somatic variants, summary figures and code is sufficient to reproduce analyzes and is available via figshare [https://figshare.com/projects/Hallmarks_of_Copy_Number_Alteration-Quiet_Oral_Cavity_Tumors/193910][68–70]. Hallmarks of Copy Number Alteration-Quiet Oral Cavity Tumors (figshare.com).

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

## Acknowledgements

The authors wish to thank all patients who participated in this study, Steven M. Mes PhD for help with and design of the MLPA assay, Widad Rifi MSc for contribution with probe design and optimization and technical assistance, Dennis N.L.M. Nijenhuis MSc for help with the multiplex immunohistochemistry, Leon Wils MSc for help with the cancer genome atlas mutational analysis, Marijke Stigter-van Walsum for help with targeted sequencing, Microscopy and Cytometry Core Facility, Amsterdam UMC for facilitating the Vectra Polaris (Akoya), clinicians and nurses from the department of Otolaryngology-head and neck surgery from Amsterdam UMC for support with tissue collection, Cancer Center Amsterdam (CCA) for financially support of this work (PV 19/02).

## Author contributions

Data collection: T.M., I.H.N., Avd.L., K.J.T.G., S.H.G. Patient database: I.H.N. Data analyzes: T.M., Avd.L., K.J.T.G., A.B., M.Avd.W., J.B.P. Visualization: T.M. Scoring of tumor and histological parameters: L.A.N.P., E.B. Providing M.L.P.A. probes: S.S., L.A. Supervision: Rvd.V., J.B.P., R.H.B. Funding acquisition: C.R.L., R.H.B., Rvd.V. Writing of the original draft: T.M. Writing review and editing: J.B.P., R.H.B., Rvd.V. Read and approved the final version of the manuscript: T.M., I.H.N., Avd.L., K.J.T.G., A.B., S.H.G., R.J.Bd.J., L.A., S.S., M.Avd.W., L.A.N.P., E.B., Rvd.V., C.R.L., J.B.P., R.H.B.

## Competing interests

The authors declare no competing interests.
