## [Peer Review File · Nature Communications]

REVIEWER COMMENTS

Reviewer #1 (Remarks to the Author): Expert in head and neck cancers, tumour microenvironment and immune microenvironment

Previous studies described a subset of oral cavity cancers (Oral cavity squamous cell carcinoma or OSCC) that have low copy number alterations (CNA) and fewer mutations than other OSCCs. However, a drawback of these studies is that the contribution of the stromal component was not considered in the analysis. In the study “Hallmarks of a Genomically Distinct Subclass of Head 1 and Neck Cancer”, Muijlwijk et al, perform an in depth analysis of the CNA-quiet and CNA-other groups relative to histopathology, CNA, and mutations in 29 genes in the previously used TCGA (the Cancer Genome Atlas) dataset and a separate multi-institutional dataset from The Netherlands. They also investigated the immune infiltrate. This is a very interesting study that is of potential clinical relevance. The following comments should be addressed.

Since cancers are an accumulation of genetic lesions, it is unclear why mutations would be a rare event in CNA-quiet HNSCC. These may have a different mutational profile from other HNSCC. Therefore, use of the 29 most commonly mutated oncogenes in all HNSCCs (weighted strongly towards the 10x more common CNA-other tumors), may not capture the mutational profile of CNA-quiet tumors. Since CNA-quiet tumors are only a subset of all HNSCCs, perhaps screening for some of the rarer mutations detected in large sequencing studies, would be a more appropriate choice.

Given the disparity in pathology between CNA-quiet and CNA-other (differentiation, invasive pattern etc.), please show representative histopathology to highlight these differences. Also include a pattern of invasion score to highlight the variation in invasive pattern rather than a qualitative assessment.

Also, how does the histopathology from the two CNA-groups in the Netherlands sample compare to the histopathology of the TCGA samples that were called on CNA and mutations alone? Histopathology for these samples is available.

Tumors can vary in invasive growth patterns. Were multiple areas sampled within a tumor? Were CNA-quiet and CNA-other areas identified in the same tumor? Could one CNA-quiet and CNA-other be different stages in tumor evolution? This is relevant because the former are early stage and the latter are usually late stage tumors.

“Tumor purity and ploidy were estimated based on copy number data and in case no CNAs were present, combined with mutation data.” How can you rule out that some of these mutations were in surface, dysplastic epithelium rather than cancer epithelium?

Fig. 4h: Given that CNA-quiet and CNA-other are typically early and late stage tumors, respectively, the disparity in survival may be a function of tumor stage rather than CNA and mutation status. If analysis was restricted to early OR late stage lesions, would there be a difference in survival between CNA-quiet and CNA-other?

Fig. 3A, the legend is incorrect. The CNA-quiet are on the left, not right as indicated.

Fig. 5: The T helper cells (CD45+, CD8-) would include Tregs. Why not quantify CD45+, CD4+, FoxP3- instead?

Fig. 5h, right panel: The sample has a core missing in the tumor center. Wouldn't this impact analysis?

It is unclear what Fig. 5A represents – CNA-quiet or CNA-other-. Besides, oral cavity SCC are rarely papillary lesions, so Fig. 5A would not be very representative of the pathology of this group of tumors. This is important because what is shown as the “invasive margin” on the left of that panel, could be surface epithelium, which would alter data quantification and interpretation; note that CD44v6 stains normal oral epithelium.

Methods, Multiplex immunohistochemistry. Why was formalin-fixed, paraffin-embedded tissue fixed again in formalin after deparaffinization?

Reviewer #2 (Remarks to the Author): Expert in head and neck cancer genomics and therapy

Muijlwijk et al.

Hallmarks of a Genomically Distinct Subclass of Head and Neck Cancer

This manuscript details the identification of what the authors deem a new molecular subclass of HPV-negative head and neck squamous cell carcinoma with a low amount of copy number alterations (CNAs). Termed “CNA-quiet”, such tumors were classified from a re-evaluation of the TCGA HNSC legacy cohort, as well as from the in-depth analysis of a large (900) HPV-negative patient cohort across two medical centers in the Netherlands. CNA-quiet tumors were classified as having copy number alterations below the statistically defined fraction genome altered (FGA) of < 0.20 . For TCGA this included < 20 called altered chromosomal segments; for the multi-center cohort, six or less altered segments through more rigorous genomic interrogation. In addition, the HPV-negative subtype was found to have few TP53 mutations and frequent mutation of CASP8 and HRAS. Additional pathological characteristics of note included a more well-differentiated tumor morphology, greater frequency in the oral cavity, lack of immunosuppression, more frequent occurrence in older females and better overall survival than HPV-negative tumors with CNAs above the 0.20 FGA threshold. The authors conclude that this group of patients represents a new molecular subtype of HPV-negative HNSCC previously missed or not emphasized by the analysis conducted by TCGA and others that may have clinical ramifications in an aging population.

The study was conducted with excellent rigor at all informatic and experimental levels, with mostly valid justification for re-examining the TCGA cohort based on concerns about stromal contamination leading to false positive identification of quiet tumors due to dilution of the overall CNA signal. This issue was

directly dealt with in the multi-center cohort utilizing amplification of chromosomal segments with multiple probes designed to target the most frequent CNAs in HPV-negative HNSCC (termed multiplex ligation dependent probe amplification) to directly assess CNA abundance. Combined with low coverage genome sequencing, the pipeline for determining quiet tumors is rational and clear. Importantly, while comparison of the quiet cohort from the re-evaluated TCGA to the multi-center group was largely congruent and in alignment with past findings, the more in-depth analysis of the quiet tumors from the multi-center study identified key aspects of the proposed quiet subclass, namely high levels of mutation of the TERT promoter, RAC1 and PIK3CA in addition to the aforementioned high CASP8, HRAS and lower TP53 mutation rates.

There is a lot of good, useful information provided in the study as far as it goes. The firm validation of a quiet HPV-negative subtype is of value. Further refinement and consideration of additional parameters have the potential to make the study of greater utility at the research and clinical levels as noted below.

1. As the authors note, a “quiet” subgroup of HPV-negative HNSCC has been previously recognized by the TCGA study, including the lack of p53 alterations, higher CASP8 and HRAS along with many (certainly not all) of the clinical parameters (ref 8 in the manuscript). As such, while the reassessment of the TCGA cohort was warranted, well conducted and backed up by the extensive work in the multi-center cohort, the only real novel genomic finding was the increase in TERT promoter mutation. While certainly an important finding in its own right, the results overall do not offer much more additional information (molecular, pathological or clinical) from what was already known or inferred from HPV-negative disease containing low CNAs.

2. Analyzed tumors were either from the TCGA database or exclusively from two centers in the Netherlands (the source that comprised the multi-center cohort). It is not clear if the quiet subtype was evaluated in the context of molecular data from other diverse genetic populations that traditionally have high HPV-negative disease. Recognizing the barriers to acquiring such samples and/or database access, such an analysis would be ideal in order to make universal claims about the identification of a new molecular subtype across all HPV-negative HNSCC.

3. While the quiet tumor subgroup does have fewer CNAs than the so-called CNA-other HPV-negative subgroups, aside from PIK3CA, HPV-positive tumors share many (certainly not all) of the same key molecular characteristics, along with most of the pathological and clinical aspects reported for quiet HPV-negative tumors (a lot of WT p53, non smokers, well differentiated, better survival, immune hot, etc.). These combined shared characteristics raise some obvious points worthy of investigation within the existing datasets, as well as commented on in the Discussion at the least:

- a. How does survival of HPV-positive patients (patients confirmed to exist in the TCGA legacy database; unknown with the multi-center group) compare with the HPV-negative quiet cohort?

- b. Importantly, detailed treatment histories of the European patients should be known. Can the treatment be teased out to determine if the better responders from the quiet cohort had the same treatment as the HPV-negative, or HPV-positive patients? This reviewer recognizes that treatment between quiet and CNA-other HPV-negative patients could very well have been the same since they

would use the identical standard of care approach. However, the idea here is if these patients are CNA quiet, can they be candidates for de-escalation as being considered and conducted with HPV-positive patients? Did such patients need full dose chemo or reduced fractionation to cure/mitigate their disease compared to CNA-other patients? This is where the study can provide some real meaningful clinical insight.

c. The mention of any patients treated in the Netherlands cohort with immune checkpoint inhibitors is not noted or clear from the text, and would have obvious ramifications given the potential for greater immune response from PD-1 inhibition as noted by the authors.

d. While cetuximab is falling out of favor as treatment for recurrent and metastatic disease due to lack of efficacy with radiation compared with traditional platinum-based therapies, it is still used and was approved during the period of time the Netherlands patient cohort was accrued. Can the authors determine if the quiet tumors respond better or worse to cetuximab treatment? This may be inferred from prior work in HPV-negative tumors where EGFR ligand abundance was deemed superior for predicting anti-EGFR monoclonal antibody inhibition (PMID: 33417831).

4. PIK3CA is 27% mutated in the quiet tumors- presuming these activating mutations, but it is not clear from the text. It is important to know this in comparison to the PIK3CA CNA and mutational profile observed by TCGA in HPV-positive cases (for example, PIK3CA is altered in 56 % out of the 36 tumors in the seminal 2015 TCGA paper (ref 8 and 9 in the manuscript)). This could be another shared characteristic between HPV-negative quiet and HPV-positive tumors, where mTOR inhibitors are a current area of clinical investigation for treatment.

5. Aside from the detailed and well-conducted multiplex IHC analysis of the immune microenvironment, how do the other components of the tumor microenvironment compare? Evaluation of stromal fibroblasts, neutrophils and endothelial cells at the minimum should be considered as it would provide additional granular data to complement the findings of this report and likely be of current and future value for translational and clinical researchers in the field.

REVIEWER COMMENTS

Reviewer #1 (Remarks to the Author): Expert in head and neck cancers, tumour microenvironment and immune microenvironment

Previous studies described a subset of oral cavity cancers (Oral cavity squamous cell carcinoma or OCSCC) that have low copy number alterations (CNA) and fewer mutations than other OSCCs. However, a drawback of these studies is that the contribution of the stromal component was not considered in the analysis. In the study “Hallmarks of a Genomically Distinct Subclass of Head and Neck Cancer”, Muijlwijk et al, perform an in depth analysis of the CNA-quiet and CNA-other groups relative to histopathology, CNA, and mutations in 29 genes in the previously used TCGA (the Cancer Genome Atlas) dataset and a separate multi-institutional dataset from The Netherlands. They also investigated the immune infiltrate. This is a very interesting study that is of potential clinical relevance. The following comments should be addressed.

We appreciate the positive comments of the reviewer.

1. Since cancers are an accumulation of genetic lesions, it is unclear why mutations would be a rare event in CNA-quiet HNSCC. These may have a different mutational profile from other HNSCC. Therefore, use of the 29 most commonly mutated oncogenes in all HNSCCs (weighted strongly towards the 10x more common CNA-other tumors), may not capture the mutational profile of CNA-quiet tumors. Since CNA-quiet tumors are only a subset of all HNSCCs, perhaps screening for some of the rarer mutations detected in large sequencing studies, would be a more appropriate choice.
 - *The reviewer raises a valid point. We therefore screened for rare mutations using the cancer genome atlas (TCGA) cohort, which provided mutation data of 15,881 genes. We compared the presence of mutations in those 15,881 genes in the CNA-other (n=339) and CNA-quiet (n=70) HNSCC from the TCGA cohort. We found next to CASP8 and HRAS in CNA-quiet tumors and enrichment of TP53 mutations in CNA-other tumors, no additional mutations enriched.*
 - *The results have been added to the revised manuscript as Supplementary Table 7, and visualized in the form of an oncoplot and onco-bar (Fig. R1, Supplementary Fig. 8).*
 - *The following text has been added to the Results of the revised manuscript:
“The panel of 29 genes was based on most common mutations in HNSCC in general. We examined the enrichment of less frequently occurring mutations in CNA-quiet tumors using TCGA data, which provides mutational data of 15,881 genes (Supplementary Fig. 8, Supplementary Table 7). We did find other mutations in CNA-quiet HNSCC, but after false discovery rate correction none remained other than CASP8, HRAS and TP53.”*

Fig. R1 | Screening for differential mutations between 339 copy number alteration (CNA)-other versus 70 CNA-quiet head and neck squamous cell carcinomas (HNSCCs). Mutations of 15,881 genes in the 409 HPV-negative HNSCCs of the cancer genome atlas (TCGA) were analyzed. Results are listed in Supplementary Table 7. **a**, OncoPrint of somatic variants of 20 most frequently mutated genes in 409 HPV-negative HNSCC samples. 339 CNA-other on the left (fraction genome altered (FGA) ≥ 0.20 and/or ≥ 20 called segments) and 70 CNA-quiet HNSCC on the right (FGA < 0.20 and < 20 called segments). Colors represent somatic variant classifications as listed in Supplementary Table 8. **b**, Fisher's exact test to compare gene mutation frequency between CNA-other and -quiet HNSCC. Raw as well as adjusted p-values (using Benjamini-Hochberg false discovery rate correction) are displayed; ten genes with lowest raw p-values are displayed.

2. Given the disparity in pathology between CNA-quiet and CNA-other (differentiation, invasive pattern etc.), please show representative histopathology to highlight these differences. Also include a pattern of invasion score to highlight the variation in invasive pattern rather than a qualitative assessment.

- We followed the comment of the reviewer and performed a comparative analysis between CNA-quiet and CNA-other in which an experienced head and neck cancer pathologist scored the pattern of invasion (POI), as reported by Heerema et al (1), in 26 CNA-quiet and 52 T-stage matched CNA-other OCSCC cases.
- Fig. R2 has been added as Fig. 5f and POI to Table 1 | Patient and tumor characteristics of the oral cavity SCC cohort in the revised manuscript (see Table R1).
- The following text has been added to the Results of the revised manuscript:

“Additionally, we performed a comparative analysis in which pattern of invasion (POI), as reported by Heerema *et al.* (18), was scored for 26 CNA-quiet and 52 CNA-other OCSCC. POI evidently differed between the groups, with CNA-quiet OCSCC solely being scored as POI-1, -2, or -3, while CNA-other was enriched for POI-4 and -5 ($p < .001$ Chi-squared test, Fig. 5d-f).”

- Representative histopathology images of CNA-quiet and CNA-other OCSCC (Fig. R3 and R4) have been added as Supplementary Fig. 11 and 12 in the revised manuscript.

Fig. R2 | Comparison of pattern of invasion (POI) between copy number alteration (CNA)-quiet and -other oral cavity squamous cell carcinoma (OCSCC). In a comparative study the POI of 26 CNA-quiet OCSCC was analyzed and compared with 52 T-stage matched CNA-other OCSCC. Relative frequencies and number of patients are depicted in the bar graph. POI was scored as reported by Heerema *et al.* (1). p -value obtained using a Chi-squared test by comparing POI1-3 with POI4-5. Unknowns were excluded from the analysis.

Table R1 | Patient and tumor characteristics of the oral cavity SCC cohort

Characteristics	Patients with OCSCC, No. (%)		
	CNA-other	CNA-quiet	
Patients	729 (90.9%)	73 (9.1%)	
Age at diagnosis, mean (range)	64 (21-92)	70 (27-91)	<.001 ^a
Sex			<.001 ^b
Female	288 (39.5%)	44 (60.3%)	
Male	441 (60.5%)	29 (39.7%)	
Tobacco use			<.001 ^b
Never	151 (20.7%)	31 (42.5%)	
Former	208 (28.5%)	26 (35.6%)	
Current	369 (50.6%)	16 (21.9%)	
Alcohol use			<.001 ^b
Never	142 (19.5%)	28 (38.4%)	
Former	80 (11.0%)	3 (4.1%)	
Current	506 (69.5%)	42 (57.5%)	
Anatomical subsite			<.001 ^b
Cheek mucosa	43 (5.9%)	15 (20.5%)	
Floor of mouth	216 (29.6%)	6 (8.2%)	
Hard palate	6 (0.8%)	2 (2.7%)	
Mobile tongue	330 (45.3%)	25 (34.2%)	
Retromolar trigone	45 (6.2%)	2 (2.7%)	
Vestibule of mouth	89 (12.2%)	23 (31.5%)	
Dental status			.27 ^b
Dentate	309 (42.4%)	27 (37.0%)	
Edentate	235 (32.2%)	28 (38.4%)	
Unknown	185 (25.4%)	18 (2.52%)	
T-stage			.48 ^b
T1	248 (34.0%)	30 (41.1%)	

T2	204 (28.0%)	23 (31.5%)	
T3	114 (15.6%)	7 (9.6%)	
T4a	149 (20.4%)	12 (16.4%)	
T4b	14 (1.9%)	1 (1.4%)	
N-stage			.006^b
N0	451 (61.9%)	60 (82.2%)	
N1	93 (12.8%)	3 (4.1%)	
N2a	23 (3.2%)	4 (5.5%)	
N2b	51 (7.0%)	3 (4.1%)	
N2c	25 (3.4%)	1 (1.4%)	
N3b	86 (11.7%)	2 (2.7%)	
Disease-stage			.009^b
I	209 (28.7%)	26 (35.6%)	
II	124 (17.0%)	21 (28.8%)	
III	121 (16.6%)	6 (8.2%)	
IVA	176 (24.1%)	18 (24.7%)	
IVB	97 (13.5%)	2 (2.7%)	
IVC	2 (0.3%)	0 (0%)	
Tissue invasion			.66^b
Bone	77 (10.6%)	11 (15.1%)	
Lamina propria	335 (46.0%)	36 (49.3%)	
Muscle	128 (17.6%)	10 (13.7%)	
Skin	1 (0.1%)	0 (0%)	
Submucosa	17 (2.3%)	3 (4.1%)	
Unknown	171 (23.5%)	13 (17.8%)	
Invasion pattern			<.001^b
Cohesive	254 (34.8%)	52 (71.2%)	
Non-cohesive	407 (55.8%)	12 (16.4%)	
Unknown	68 (9.3%)	9 (12.3%)	
POI category			<.001^b
POI-1-3	28 (3.8%)	26 (28.0%)	
POI-4-5	24 (3.3%)	0 (0%)	
Unknown	677 (92.9%)	47 (64.4%)	
Differentiation grade			<.001^b
Well-differentiated	87 (11.9%)	31 (42.5%)	
Moderately-differentiated	403 (55.3%)	32 (43.8%)	
Poorly-differentiated	178 (24.4%)	5 (6.8%)	
Unknown	61 (8.3%)	5 (6.8%)	
Surgical margins			.67^b
Clear (>5mm)	291 (39.9%)	33 (45.2%)	
Close (1-5mm)	191 (26.2%)	21 (28.8%)	
Involved (<1mm)	133 (18.2%)	12 (16.4%)	
Unknown	114 (15.6%)	7 (1%)	
Extranodal extensions (pathological)			.61^b
Yes	98 (13.4%)	6 (8.2%)	
No	428 (58.7%)	33 (45.2%)	
Unknown	203 (27.8%)	34 (46.6%)	
Treatment type			.34^b
Radiotherapy (RT)	18 (5.1%)	1 (4.2%)	
Chemoradiotherapy (CRT)	14 (4.0%)	0 (0%)	
Surgery	78 (22.0%)	9 (37.5%)	
Surgery + RT	210 (59.3%)	14 (58.3%)	
Surgery + CRT	29 (8.2%)	0 (0%)	
Other	5 (1.4%)	0 (0%)	

Abbreviations: OCSCC, oral cavity squamous cell carcinoma; T, N and disease stage according to TNM classification 8th edition, 2017. Pathological stage was used when available; when the patient was not treated with surgery, clinical stage was used. Pattern of invasion (POI) as reported by Heerema et al. (1). For treatment type, only advanced disease-staged (III-IV) tumors with curative intent were included. ^a p-value obtained by Mann-Whitney test, ^b p-value obtained by Chi-squared test. Unknowns were excluded from the analysis.

Fig. R3 | Representative histopathology of CNA-quiet oral cavity squamous cell carcinoma (OCSCC). **a**, Sample 215: T1, cohesive, pattern of invasion (POI)-1, moderately-differentiated CNA-quiet OCSCC. **b**, Sample 412: T4a, cohesive, POI-1, well-differentiated CNA-quiet OCSCC. POI was scored by a trained pathologist as reported by Heerema *et al.* (1).

Fig. R4 | Representative histopathology of CNA-other oral cavity squamous cell carcinoma (OCSCC). **a**, Sample 433: T1, non-cohesive, pattern of invasion (POI)-4, moderately-differentiated CNA-other OCSCC. **b**, Sample 352: T4a, non-cohesive, POI-4, moderately-differentiated CNA-other OCSCC. **c**, Sample 259: T1, non-cohesive, POI-4, poorly-differentiated CNA-other OCSCC. POI was scored by a trained pathologist as reported by Heerema *et al.* (1).

3. Also, how does the histopathology from the two CNA-groups in the Netherlands sample compare to the histopathology of the TCGA samples that were called on CNA and mutations alone? Histopathology for these samples is available.

- *In the TCGA, in line with our findings, CNA-quiet tumors were enriched for well-differentiated tumors ($p < .001$, Fig. R5).*
- *The following text has been added to the Results of the revised manuscript:
"As histological grade is also available in the TCGA, we were able to confirm the enrichment of well-differentiated tumors in the CNA-quiet group (30% well-differentiated in CNA-quiet versus 10% in CNA-other group, $p < .001$)."*

Fig. R5 | Comparison of pattern of histological differentiation grade between copy number alteration (CNA)-quiet ($n=69$) and -other ($n=332$) HNSCC from the cancer genome atlas (TCGA). Relative frequencies and number of patients. p -value obtained using a Chi-squared test. Unknowns were excluded from the analysis.

4. Tumors can vary in invasive growth patterns. Were multiple areas sampled within a tumor? Were CNA-quiet and CNA-other areas identified in the same tumor? Could one CNA-quiet and CNA-other be different stages in tumor evolution? This is relevant because the former are early stage and the latter are usually late stage tumors.

- *Interesting and relevant comment of the reviewer. Previously we published on intratumor genetic heterogeneity by multiregion sequencing a set of eleven oral cancers sampled at multiple locations (Pierik et al. Radiotherapy & Oncology 2024 (2)). Ten out of eleven tumors studied were CNA-other and all biopsies remained CNA-other. One case was CNA-quiet and in all biopsies it remained CNA-quiet. To extend the latter group, we analyzed multiple biopsies from nine CNA-quiet cases to determine potential CNA heterogeneity. We performed low coverage whole genome sequencing (lcWGS) as well as targeted sequencing on the isolated DNA.*
- *The results can be found in Fig. R6. In the revised manuscript we presented the profiles of three representative tumors and added those as main Fig. 3. The remaining tumors were added as supplemental Fig. 7.*

- The following text has been added to the Methods section of the revised manuscript:

“CNA intratumor heterogeneity in multiple biopsies
 Multiple tumor core biopsies were punched in different blocks for nine CNA-quiet OCSCC, according to aforementioned methods used for the entire cohort. After measuring DNA yield, lcWGS and targeted sequencing was performed according KAPA HyperCap FFPE DNA Workflow v1.1 with nine PCR cycles to obtain copy number plots and somatic mutations present in the tumor core biopsies to test for CNA intratumor heterogeneity.”
- In addition, the following text has been added to the Results of the revised manuscript:

“Multiple CNA-quiet biopsies demonstrates minor CNA heterogeneity
 Previously we tested CNA and mutation intratumor heterogeneity in oral cancer by genetic analysis of multiple biopsies (2). Generally, CNAs and mutations in the cancer driver genes were very comparable, although some heterogeneity was observed. One out of the eleven studied cases was CNA-quiet and it remained CNA-quiet in the two biopsies studied. To extend this group, we tested multiple biopsies of nine CNA-quiet OCSCC. Three examples are shown in Fig. 3 and the remaining cases in Supplementary Fig. 7. Despite some minor changes between biopsies, eight out of nine CNA-quiet OCSCC remained CNA-quiet in all biopsies. One case showed somewhat more CNA heterogeneity, as one out of three biopsies was classified as CNA-other with an FGA just above 0.20 (Fig. R6d).”

Fig. R6 | Multiregion sequencing of nine CNA-quiet oral cavity SCC (OCSCC) to analyze CNA intratumor heterogeneity. a, Fraction genome altered (FGA, y-axis) across multiple biopsies from CNA-quiet OCSCCs (x-axis). b-j, Copy number plots of sample b, 264, c, 177 d, 422, e, 132, f, 447, g, 36, h, 208, i, 73 and j, 459. Called segments in red, as output from low coverage whole genome sequencing (lcWGS).

5. “Tumor purity and ploidy were estimated based on copy number data and in case no CNAs were present, combined with mutation data.” How can you rule out that some of these mutations were in surface, dysplastic epithelium rather than cancer epithelium?

- DNA was isolated from FFPE punch biopsies taken in the tumor center. This was indicated in the method section of the manuscript as follows:
 - “A core biopsy of 1 mm diameter was taken from the tumor area of FFPE specimen by guidance of the corresponding hematoxylin & eosin (H&E)-slide using a plunger system (Kai Europe GmbH, Solingen, Germany). Genomic DNA from the core biopsies was isolated using the NucliSENS easyMag (bioMérieux, Marcy-l'Étoile, France).”
- With the guidance of the H&E slide, the tumor center was identified with help of a trained head and neck cancer pathologist to make sure the punch was taken in the center of the tumor. An example of this 1mm punch biopsy can be noticed in Fig. R8.

Fig. R8 | Multiplex immunohistochemistry (IHC) image with punch taken in tumor center for DNA isolation followed by copy number alteration (CNA) classification.

6. Fig. 4h: Given that CNA-quiet and CNA-other are typically early and late stage tumors, respectively, the disparity in survival may be a function of tumor stage rather than CNA and mutation status. If analysis was restricted to early OR late stage lesions, would there be a difference in survival between CNA-quiet and CNA-other?

- As the reviewer indicates, early disease-stage (Stage I-II) was more frequent in CNA-quiet OCSCC (64% versus 46%) while late disease-stage (Stage III-IV) was more common for CNA-other OCSCC (54% versus 36%, $p=.002$, Fig. R9 and main Fig 5g).

- As seen in the multivariate analysis, early disease-stage contributed to an improved overall survival (Table R2 and main Table 2). As mentioned in the main text, CNA-quiet status is a predictor even when taking disease-stage into account.
- We analyzed early (I-II) and late disease-stage (III-IV) separately (Fig. R10 – added as Supplemental Fig. 13a-b in revised manuscript).
- The difference in prognosis between CNA-quiet and -other was especially noticed in late disease-stage. When the analysis was restricted to early stage lesions, we did not find a significant difference between CNA-quiet and CNA-other tumors ($p=.21$). In advanced disease-stage, CNA-other tumors particularly exhibited poor performance compared to CNA-quiet tumors ($p=.003$). This is not unexpected: the prognosis of early stage disease is very favorable in general, and the effectiveness of surgery at this stage does not leave room for tumor biological factors to impact survival.
- The following underlined text has been added to the Results of the revised manuscript:

“Importantly, CNA-quiet OCSCC exhibited markedly better overall survival compared to CNA-other OCSCC (HR 0.44, $p=.002$, Fig. 4h). Of note, for the survival analysis, patients treated with palliative intent ($n=4$ CNA-quiet and $n=49$ CNA-other) were excluded, leaving only patients treated with curative intent ($n=69$ CNA-quiet and $n=680$ CNA-other). In a multivariate model, including all parameters that significantly differed between CNA-quiet and CNA-other, the CNA status remained an independent prognostic variable (Table 2). To disentangle disease-stage, that differed between CNA-quiet and CNA-other OCSCC groups, on survival, we conducted separate survival analyses for early and late disease-stages (Supplementary Fig. 13a-b). When the analysis was restricted to early-stage lesions, no significant difference was found ($p=.21$) for CNA-quiet versus CNA-other. However, the more favorable prognosis of CNA-quiet OCSCC was particularly pronounced in late-stage disease (III-IV, $p=.003$).”

Fig. R9 | Comparison of disease-stage between copy number alteration (CNA)-quiet ($n=70$) and -other ($n=729$) HNSCC. Relative frequencies and number of patients. Disease-stage according to TNM classification 8th edition, 2017. Pathological stage was used when available; when the patient was not treated with surgery, p -value obtained using a Chi-squared test. Unknowns were excluded from the analysis.

Table R2 | Multivariate analysis of the multicenter oral cavity SCC cohort

Final classification	Hazard ratio	p-value
CNA-other	Reference	Reference
CNA-quiet	0.55	.04
Age	1.03	<.001
Sex		
Female	Reference	Reference
Male	1.25	.10
Tobacco use		
Never	0.64	.02
Former	0.91	.49
Current	Reference	Reference
Alcohol use		
Never	1.20	.32
Former	1.28	.20
Current	Reference	Reference
Anatomical subsite		
Cheek mucosa	Reference	Reference
Floor of mouth	0.78	.38
Hard palate	0.93	.90
Mobile tongue	0.85	.53
Retromolar trigone	0.74	.37
Vestibule of mouth	0.97	.93
N-stage		
N0	Reference	Reference
N1	1.07	.76
N2a	1.49	.25
N2b	1.51	.15
N2c	2.01	.05
N3b	2.56	.08
Disease-stage		
I	Reference	Reference
II	1.40	.12
III	2.22	.001
IVA	2.25	.002
IVB	2.18	.15
Invasion pattern		
Cohesive	Reference	Reference
Non-cohesive	1.22	.17
Unknown	1.53	.12
Differentiation grade		
Well-differentiated	0.72	.14
Moderately-differentiated	Reference	Reference
Poorly-differentiated	1.11	.44
Unknown	1.03	.91

P-values obtained by logrank test on Cox proportional hazard model. Only patients with curative treatment intent (n=69 CNA-quiet and n=680 CNA-other) were included for the analysis; patients with palliative intent (n=4 CNA-quiet and n=49 CNA-other) were excluded.

a**b**
Fig. R10 | Survival analyses early and late disease-stage separately. Kaplan-Meier curve for CNA-quiet and -other OCSCC with p-value and hazard ratio (HR) displayed, obtained by log-rank test. Patients with curative treatment intent were included in the survival analysis; patients with palliative intent were excluded. Analyses separately for **a**, early (I-II) and **b**, late (III-IV) disease-stage.

7. Fig. 3A, the legend is incorrect. The CNA-quiet are on the left, not right as indicated.
 - *We thank the reviewer for pointing this out, we have changed this in the revised manuscript.*

8. Fig. 5: The T helper cells (CD45+. CD8-) would include Tregs. Why not quantify CD45+,CD4+,FoxP3- instead?
 - *We indicated that CD4+ T helper cells were identified by CD3+ CD8-. This information was incomplete, we also selected for FoxP3- cells when quantifying CD4+ T helper cells. We apologize for any confusion, we changed it in the revised manuscript to CD4+ T helper cells (CD3+ CD8- FoxP3-).*
 - *CD45 and CD4 were not included in the seven-color multiplex immunohistochemistry panel and therefore could not be used for the annotation of CD4+ T helper cells.*

9. Fig. 5h, right panel: The sample has a core missing in the tumor center. Wouldn't this impact analysis?
 - *There was a punch biopsy taken at the tumor core for copy number alteration (CNA) classification. As mentioned in point 5, we took 1mm punch biopsies at the tumor center, followed by DNA isolation, multiplex ligation-dependent probe amplification (MLPA), low coverage whole genome sequencing (lcWGS) and targeted enrichment sequencing in order to classify a tumor as CNA-quiet or CNA-other.*

- For multiplex immunohistochemistry analysis, the density of immune cells was calculated by dividing the cell count by the tumor area (in mm^2). As the punch biopsy was not included as tumor area, this did not impact analysis.

10. It is unclear what Fig. 5A represents – CNA-quiet or CNA-other. Besides, oral cavity SCC are rarely papillary lesions, so Fig. 5A would not be very representative of the pathology of this group of tumors. This is important because what is shown as the “invasive margin” on the left of that panel, could be surface epithelium, which would alter data quantification and interpretation; note that CD44v6 stains normal oral epithelium.

- We are aware that CD44v6 is not a specific tumor marker and also stains adjacent normal and dysplastic tissue when present. CD44v6 was not used to distinguish tumor from adjacent normal epithelium. A trained head and neck cancer pathologist annotated the tumor area and when present adjacent normal epithelium, dysplastic and tumor tissue.
- See Fig. R11b for the complete image of this particular tumor. Only tumor area (tumor center + inner margin + outer margin) was analyzed. Dysplastic and normal adjacent tissue were excluded.
- We agree with the reviewer that the tumor depicted in Fig. 5a (see Fig. R11a) is not the most representative for this group of tumor. Therefore, we have selected another example for Fig. 5a (Fig. R11c).

Fig. R11 | Multiplex immunohistochemistry (IHC) images. **a**, Image in previous manuscript used in Fig. 5a as representative image. **b**, Same tumor but now the whole image is presented to show that adjacent normal and dysplastic tumor was annotated by a trained pathologist but was excluded for further analyses. **c**, Image which we now present in Fig. 5a (Changed to Fig. 6a), which is more representative.

11. Methods, Multiplex immunohistochemistry. Why was formalin-fixed, paraffin-embedded tissue fixed again in formalin after deparaffinization?

- *The 4 μ m sections were fixed for 2 hours in formalin on superfrost plus adhesion microscopic glass slides to increase binding to the glass. Otherwise the tissue will not adhere in the following washing and microwave retrieval steps.*
- *For clarification the following underlined text has been added to the Methods section of the revised manuscript:
"The tissue sections were fixed by immersing the slides in 10% neutral buffered formalin for two hours in order to prevent the tissue from detaching from the microscopic glass slide."*

Reviewer #2 (Remarks to the Author): Expert in head and neck cancer genomics and therapy
 This manuscript details the identification of what the authors deem a new molecular subclass of HPV-negative head and neck squamous cell carcinoma with a low amount of copy number alterations (CNAs). Termed "CNA-quiet", such tumors were classified from a re-evaluation of the TCGA HNSC legacy cohort, as well as from the in-depth analysis of a large (900) HPV-negative patient cohort across two medical centers in the Netherlands. CNA-quiet tumors were classified as having copy number alterations below the statistically defined fraction genome altered (FGA) of < 0.20. For TCGA this included < 20 called altered chromosomal segments; for the multi-center cohort, six or less altered segments through more rigorous genomic interrogation. In addition, the HPV-negative subtype was found to have few TP53 mutations and frequent mutation of CASP8 and HRAS. Additional pathological characteristics of note included a more well-differentiated tumor morphology, greater frequency in the oral cavity, lack of immunosuppression, more frequent occurrence in older females and better overall survival than HPV-negative tumors with CNAs above the 0.20 FGA threshold. The authors conclude that this group of patients represents a new molecular subtype of HPV-negative HNSCC previously missed or not emphasized by the analysis conducted by TCGA and others that may have clinical ramifications in an aging population.

The study was conducted with excellent rigor at all informatic and experimental levels, with mostly valid justification for re-examining the TCGA cohort based on concerns about stromal contamination leading to false positive identification of quiet tumors due to dilution of the overall CNA signal. This issue was directly dealt with in the multi-center cohort utilizing amplification of chromosomal segments with multiple probes designed to target the most frequent CNAs in HPV-negative HNSCC (termed multiplex ligation dependent probe amplification) to directly assess CNA abundance. Combined with low coverage genome sequencing, the pipeline for determining quiet tumors is rational and clear. Importantly, while comparison of the quiet cohort from the re-evaluated TCGA to the multi-center group was largely congruent and in alignment with past findings, the more in-depth analysis of the quiet tumors from the multi-center study identified key aspects of the proposed quiet subclass, namely high levels of mutation of the TERT promoter, RAC1 and PIK3CA in addition to the aforementioned high CASP8, HRAS and lower TP53 mutation rates. There is a lot of good, useful information provided in the study as far as it goes. The firm validation of a quiet HPV-negative subtype is of value. Further refinement and consideration of additional parameters have the potential to make the study of greater utility at the research and clinical levels as noted below.

1. As the authors note, a "quiet" subgroup of HPV-negative HNSCC has been previously recognized by the TCGA study, including the lack of p53 alterations, higher CASP8 and HRAS along with many (certainly not all) of the clinical parameters (ref 8 in the manuscript). As such, while the reassessment of the TCGA cohort was warranted, well conducted and backed

up by the extensive work in the multi-center cohort, the only real novel genomic finding was the increase in TERT promoter mutation. While certainly an important finding in its own right, the results overall do not offer much more additional information (molecular, pathological or clinical) from what was already known or inferred from HPV-negative disease containing low CNAs.

- *We thank the reviewer for the kind comments, but respectfully disagree with the aspect of novelty. Yes the CNA-quiet group had been identified by us and the TCGA, but much more rigorous methods were required to set the stage, which is all presented in this manuscript:*
 - *We provide a quantitative measure to distinguish CNA-quiet tumors for research purposes as well as for clinical application. Up till now, the CNA-quiet group was only identified by clustering analyses (3, 4), which misses more rigorous criteria to define the CNA-quiet group.*
 - *We developed analyses to exclude stromal contamination, a major problem in the previously reported studies.*
 - *We describe and present histopathological features of the CNA-quiet as they are primarily cohesive and moderately- or well-differentiated tumors with pattern of invasion (POI)-1, -2, or -3.*
 - *Also, we found that CNA-quiet OCSCC were significantly more located in cheek mucosa and vestibule of mouth while less often in floor of mouth.*
 - *We are the first study to perform spatial immunological analysis on CNA-quiet OCSCC. As highlighted in the discussion:*
“It has been reported that many CNAs in the genome negatively correlated with interferon- γ signaling (5-7), the expression of T cell markers (6, 7), B cell and NK cell markers (7), as well as immune cell infiltration (6, 8). Of note, those studies were merely based on deconvolution analyses from gene expression data (5-9), and generally not on spatial analysis of the TIME.”
 - *Lastly, as an addition to the comment of reviewer 1 we provided extra data on the intratumor genetic heterogeneity of the CNA-quiet subgroup.*
- *All these data really set the stage for the CNA-quiet subgroup of oral cancers that typically occur in aged female patients with limited to minimal carcinogen exposure.*

2. Analyzed tumors were either from the TCGA database or exclusively from two centers in the Netherlands (the source that comprised the multi-center cohort). It is not clear if the quiet subtype was evaluated in the context of molecular data from other diverse genetic populations that traditionally have high HPV-negative disease. Recognizing the barriers to acquiring such samples and/or database access, such an analysis would be ideal in order to make universal claims about the identification of a new molecular subtype across all HPV-negative HNSCC.

- *We agree with the reviewer that it is essential that comparable studies will be executed in other cohorts to further validate our findings. In this manuscript we show that this subgroup is really different in many aspects, and we provide the relevant tools to study them. We hope that our manuscript will be the start of analyses of this cohort over the world, also in relation to other habits such as betel nut chewing.*
3. While the quiet tumor subgroup does have fewer CNAs than the so-called CNA-other HPV-negative subgroups, aside from PIK3CA, HPV-positive tumors share many (certainly not all) of the same key molecular characteristics, along with most of the pathological and clinical aspects reported for quiet HPV-negative tumors (a lot of WT p53, nonsmokers, well differentiated, better survival, immune hot, etc.). These combined shared characteristics raise some obvious points worthy of investigation within the existing datasets, as well as commented on in the Discussion at the least:
- a. How does survival of HPV-positive patients (patients confirmed to exist in the TCGA legacy database; unknown with the multi-center group) compare with the HPV-negative quiet cohort?
 - *This is an interesting point for consideration brought up by the reviewer. However, HPV-positive HNSCC typically occur in the oropharynx. Both HPV-negative and HPV-positive oropharynx SCC (OPSCC) are treated differently compared to OCSCC (definitive chemoradiotherapy instead of upfront surgery), making these groups difficult to compare with respect to outcome. Moreover, HPV-positive OPSCC have a well-established other etiology, and are widely accepted as separate disease entity but should for prognosis always be compared with HPV-negative OPSCC. This all makes relevant comparisons between HPV-positive OPSCC and HPV-negative OCSCC difficult, and even confusing in our view.*
 - *However we decided to perform a comparative analysis with HPV-positive OCSCC. As reported by Nauta et al. 2021 (10), only a very small percentage of OCSCC (2.2%) are HPV-positive. Moreover, a more favorable prognosis was not reported for HPV-positive OCSCC. As we utilized the same OCSCC cohort as reported by Nauta et al. 2021 (10), we selected, next to the 802 HPV-negative OCSCC, also the HPV-positive OCSCC cases (n=21) and compared overall survival between HPV-positive OCSCC and CNA-quiet HPV-negative OCSCC. There was no difference in survival (Fig. R12 – added as Supplemental Fig. 13c).*
 - *In summary, while for OPSCC the difference between HPV-negative and HPV-positive disease is critical, for OCSCC, HPV involvement is of no relevance. For OCSCC, the distinction between CNA-quiet and CNA-other is much more relevant. While superficially comparable due to some shared characteristics, differences in both their etiology and prognostic characteristics predominate.*
 - *The following text has been added to the Discussion of the revised manuscript: “HPV-negative CNA-quiet OCSCC and HPV-positive HNSCC have some shared clinical (improved prognosis, generally non-smoking) and histological (well-differentiated) characteristics. However, HPV-positive HNSCC typically occur in the oropharynx. While we acknowledge some similarities, HPV-positive*

OPSCC is seen as separate disease entity compared to HPV-negative OPSCC. Within OCSCC, HPV does not play a prognostic role, as reported by Nauta et al. (46). We analyzed the survival of HPV-positive OCSCC, CNA-other and CNA-quiet OCSCC and only CNA-quiet OCSCC showed a more favorable outcome (Supplementary Fig. 13c). Taken together, while for OPSCC the difference between HPV-negative and HPV-positive is critical, for OCSCC, the distinction between CNA-quiet and CNA-other is crucial.”

Fig. R12 | Survival analysis including HPV-positive oral cavity SCC. Kaplan-Meier curve for CNA-quiet and CNA-other HPV-negative OCSCC and HPV-positive OCSCC with p-value and hazard ratio (HR) displayed, obtained by log-rank test. Patients with curative treatment intent were included in the survival analysis; patients with palliative intent were excluded.

- b. Importantly, detailed treatment histories of the European patients should be known. Can the treatment be teased out to determine if the better responders from the quiet cohort had the same treatment as the HPV-negative, or HPV-positive patients? This reviewer recognizes that treatment between quiet and CNA-other HPV-negative patients could very well have been the same since they would use the identical standard of care approach. However, the idea here is if these patients are CNA quiet, can they be candidates for de-escalation as being considered and conducted with HPV-positive patients? Did such patients need full dose chemo or reduced fractionation to cure/mitigate their disease compared to CNA-other patients? This is where the study can provide some real meaningful clinical insight.
 - This is an interesting point for consideration. We added treatment type in Table 1 as well as in Supplementary Table 2 (column CB, CD and CE).

- Treatment given to the CNA-quiet group is indeed comparable as for the CNA-other group, typically surgery with or without post-operative (chemo)radiotherapy. Obviously, it is totally different from HPV-positive OPSCC, which are treated with definitive (chemo)radiotherapy.
- Interestingly, the advanced staged (III-IV) CNA-quiet tumors apparently did not require post-operative chemotherapy because of their relatively favorable clinical and histological characteristics, while some of the CNA-other tumors clearly required postoperative chemoradiotherapy (Fig. R13 – added as Supplementary Fig. 10e). In order to establish whether de-escalation is possible for CNA-quiet tumors, a randomized controlled trial should be executed. In this manuscript we provide the tools to define this group.
- The following text has been added to the Discussion of the revised manuscript: “Interestingly, the difference in prognosis between CNA-quiet and -other OCSCC was notably accentuated during late-stage disease. Remarkably, none of the advanced disease-stage CNA-quiet OCSCC received surgery + postoperative chemoradiotherapy (Supplementary Fig. 10e), presumably due to their more favorable clinical and histological characteristics.”

Fig. R13 | Comparison of treatment type for advanced stage (III-IV) copy number alteration (CNA)-quiet (n=19) and -other (n=246) OCSCC. Relative frequencies and number of patients. Disease-stage according to TNM classification 8th edition, 2017. Pathological stage was used when available; when the patient was not treated with surgery, p-value obtained using a Chi-squared test. Unknowns were excluded from the analysis. Patients with curative treatment intent were included in the survival analysis; patients with palliative intent were excluded.

- The mention of any patients treated in the Netherlands cohort with immune checkpoint inhibitors is not noted or clear from the text, and would have obvious ramifications given the potential for greater immune response from PD-1 inhibition as noted by the authors.

- *Immune checkpoint inhibitors have been registered since 2018 for recurrent/metastatic disease. In this cohort no recurrent/metastatic HNSCC were included, and none of them were treated with immune checkpoint inhibitors in clinical trials.*
 - *Obviously it would be of great value to perform CNA classification on specifically an oral cancer cohort treated with immune checkpoint inhibitors to examine differences in response, but this requires analyses of running clinical trial populations.*
- d. While cetuximab is falling out of favor as treatment for recurrent and metastatic disease due to lack of efficacy with radiation compared with traditional platinum-based therapies, it is still used and was approved during the period of time the Netherlands patient cohort was accrued. Can the authors determine if the quiet tumors respond better or worse to cetuximab treatment? This may be inferred from prior work in HPV-negative tumors where EGFR ligand abundance was deemed superior for predicting anti-EGFR monoclonal antibody inhibition (PMID: 33417831).
- *This is an interesting point for consideration. Unfortunately, we do not have data on this, none of the patients in the current cohort were treated with cetuximab.*
4. PIK3CA is 27% mutated in the quiet tumors- presuming these activating mutations, but it is not clear from the text. It is important to know this in comparison to the PIK3CA CNA and mutational profile observed by TCGA in HPV-positive cases (for example, PIK3CA is altered in 56% out of the 36 tumors in the seminal 2015 TCGA paper (ref 8 and 9 in the manuscript)). This could be another shared characteristic between HPV-negative quiet and HPV-positive tumors, where mTOR inhibitors are a current area of clinical investigation for treatment.
- *The majority of the PIK3CA mutations reported were known pathogenic, activating mutations. Some were not annotated but frequently occurring in cancer, and only a few were unknown (detailed list of PIK3CA mutations in our cohort has been added as Supplementary Table 6 and can be found as Fig. R15 below).*
 - *PIK3CA mutations were enriched in the CNA-quiet group of our multicenter OCSCC cohort: 20 out of 73 (27%) CNA-quiet harbored a PIK3CA mutation while this was 7 out of 71 (10%) in the CNA-other group (raw $p=.02$; adjusted $p=.13$, Fig. R14 and main Fig. 4b). However, the multiple testing corrected p -values did not indicate a significant difference. Whether the CNA-quiet tumors are sensitive to mTOR inhibitors needs to be tested in the future.*

Fig. R14 | Mutational profile of 73 copy number alteration (CNA)-quiet and 71 CNA-other oral cavity SCC (OCSCC). Fisher's exact test to compare mutations between CNA-quiet and -other OCSCC. Raw as well as adjusted *p*-value (using Benjamini-Hochberg false discovery rate) are displayed; 10 genes with lowest *p*-value are displayed and ordered based on their *p*-value.

Sample	CMA classification	FGA	Called segments	Cellularity	CHROM	POS	REF	ALT	DP	AD	VAF	GENE	variantClassification	variantType	codonChange	proteinChange	clinVar	WGT	CINSG	dbSNP ID
217	CNA, other				chr3	178928010	T	C	104	21	0.20193	PKCEA	MISSENSE	SNP	c.12185-1200(T>C)	p.F430L				r1108479372
526	CNA, other				chr3	178922231	G	A	539	166	0.286846	PKCEA	MISSENSE	SNP	c.1095-1092(G>A)	p.D348H				r1104886003
668	CNA, other				chr3	178936882	C	G	612	482	0.722222	PKCEA	MISSENSE	SNP	c.1624-1626(G>C)	p.E547Q				r1104886003
677	CNA, other				chr3	178921532	A	T	641	132	0.205928	PKCEA	MISSENSE	SNP	c.1103-1035(A>T)	p.N434I				r1121913278
1131	CNA, other				chr3	178937885	A	T	42	20	0.47819	PKCEA	MISSENSE	SNP	c.1319-1341(A>T)	p.H467Y				r1121913278
1176	CNA, other				chr3	178936991	G	A	1006	409	0.406591	PKCEA	MISSENSE	SNP	c.1635-1635(G>A)	p.E548K				r1121913279
32	CNA, quiet				chr3	178936882	G	A	534	134	0.241827	PKCEA	MISSENSE	SNP	c.1058-1026(G>A)	p.E542K				r11057919938
36	CNA, quiet				chr3	178936991	G	A	190	50	0.261548	PKCEA	MISSENSE	SNP	c.1635-1635(G>A)	p.E548K				r11057919938
48	CNA, quiet				chr3	178936991	G	A	189	66	0.349206	PKCEA	MISSENSE	SNP	c.1635-1635(G>A)	p.E548K				r11057919938
177	CNA, quiet				chr3	178936882	G	A	52	20	0.384615	PKCEA	MISSENSE	SNP	c.1624-1626(G>A)	p.E542K				r1121913279
229	CNA, quiet				chr3	178936991	G	A	97	19	0.195876	PKCEA	MISSENSE	SNP	c.1635-1635(G>A)	p.E548K				r1104886003
364	CNA, quiet				chr3	178936991	G	A	97	19	0.195876	PKCEA	MISSENSE	SNP	c.1635-1635(G>A)	p.E548K				r1104886003
386	CNA, quiet				chr3	178936991	G	A	97	19	0.195876	PKCEA	MISSENSE	SNP	c.1635-1635(G>A)	p.E548K				r1104886003
411	CNA, quiet				chr3	178936991	G	A	112	56	0.188454	PKCEA	MISSENSE	SNP	c.1635-1635(G>A)	p.E548K				r1121913279
421	CNA, quiet				chr3	178936991	G	A	112	56	0.188454	PKCEA	MISSENSE	SNP	c.1635-1635(G>A)	p.E548K				r1121913279
483	CNA, quiet				chr3	178923864	A	A	48	8	0.166667	PKCEA	MISSENSE	SNP	c.1132-1134(G>A)	p.C378Y				r1104886003
488	CNA, quiet				chr3	178936991	G	A	57	17	0.289546	PKCEA	MISSENSE	SNP	c.1635-1635(G>A)	p.E548K				r1104886003
488	CNA, quiet				chr3	178936882	G	A	797	342	0.429109	PKCEA	MISSENSE	SNP	c.1624-1626(G>A)	p.E542K				r1104886003
597	CNA, quiet				chr3	178936991	G	A	799	296	0.387183	PKCEA	MISSENSE	SNP	c.1635-1635(G>A)	p.E548K				r1104886003
613	CNA, quiet				chr3	178936991	G	A	177	48	0.271188	PKCEA	MISSENSE	SNP	c.1635-1635(G>A)	p.E548K				r1104886003
743	CNA, quiet				chr3	178931944	C	C	357	82	0.229692	PKCEA	MISSENSE	SNP	c.1006-1051(G>C)	p.D1017N				r1104886003
743	CNA, quiet				chr3	178952035	A	G	263	53	0.201523	PKCEA	MISSENSE	SNP	c.1335-1341(A>G)	p.H467Y				r1104886003
761	CNA, quiet				chr3	178916755	C	C	107	32	0.299905	PKCEA	MISSENSE	SNP	c.142-144(G>C)	p.A48Q				r1104886003
796	CNA, quiet				chr3	178936991	G	A	113	12	0.106185	PKCEA	MISSENSE	SNP	c.1635-1635(G>A)	p.E548K				r1104886003
857	CNA, quiet				chr3	178938922	G	C	477	76	0.159129	PKCEA	MISSENSE	SNP	c.2164-2166(G>C)	p.R722Q				r1104886003
857	CNA, quiet				chr3	178938922	G	C	477	76	0.159129	PKCEA	MISSENSE	SNP	c.2164-2166(G>C)	p.R722Q				r1104886003
1165	CNA, quiet				chr3	178921532	A	T	676	281	0.415668	PKCEA	MISSENSE	SNP	c.1035-1035(A>T)	p.N434I				r1121913279
1185	CNA, quiet				chr3	178923224	A	A	479	83	0.193473	PKCEA	MISSENSE	SNP	c.1109-1095(G>A)	p.E305K				r1121913279

Fig. R15 | PIK3CA mutations in oral cavity SCC (OCSCC) cohort.

5. Aside from the detailed and well-conducted multiplex IHC analysis of the immune microenvironment, how do the other components of the tumor microenvironment compare? Evaluation of stromal fibroblasts, neutrophils and endothelial cells at the minimum should be considered as it would provide additional granular data to complement the findings of this report and likely be of current and future value for translational and clinical researchers in the field.
- *It would be interesting to evaluate stromal fibroblasts, neutrophils and endothelial cells. Specific multiplex immunohistochemistry (IHC) panels have to be developed to spatially investigate this, making it a study on itself and it was not the focus of our current study.*
 - *For now, we have chosen to apply the immunologically focused seven-color multiplex IHC Opal panel characterizing CD44v6+ tumor cells, CD163+ (tumor promoting) macrophages, CD19+ B cells, CD8+ T cells, CD3+ CD8- FoxP3- T cells (CD4+ T helper cells) and FoxP3+ regulatory T cells (Tregs, CD3+, CD8-).*
 - *Another interesting way to investigate the tumor microenvironment would be using single cell RNA-sequencing (scRNA-seq). We aimed to compare CNA-quiet and CNA-other in a prospective cohort using flow cytometry and scRNA-seq (11). Unfortunately, no CNA-quiet tumors were available for fresh tumor digestion. Hence this also remains an interesting approach for the future. Our current study is instrumental to show that it is worthwhile to investigate these various aspects.*

References

1. Heerema MG, Melchers LJ, Roodenburg JL, Schuurin E, de Bock GH, van der Vegt B. Reproducibility and prognostic value of pattern of invasion scoring in low-stage oral squamous cell carcinoma. *Histopathology*. 2016;68(3):388-97.
2. Pierik AS, Poell JB, Brink A, Stigter-van Walsum M, de Roest RH, Poli T, et al. Intratumor genetic heterogeneity and head and neck cancer relapse. *Radiother Oncol*. 2024;191:110087.
3. Cancer Genome Atlas N. Comprehensive genomic characterization of head and neck squamous cell carcinomas. *Nature*. 2015;517(7536):576-82.
4. Smeets SJ, Brakenhoff RH, Ylstra B, van Wieringen WN, van de Wiel MA, Leemans CR, Braakhuis BJ. Genetic classification of oral and oropharyngeal carcinomas identifies subgroups with a different prognosis. *Cell Oncol*. 2009;31(4):291-300.
5. Buccitelli C, Salgueiro L, Rowald K, Sotillo R, Mardin BR, Korbel JO. Pan-cancer analysis distinguishes transcriptional changes of aneuploidy from proliferation. *Genome Res*. 2017;27(4):501-11.
6. Mandal R, Senbabaoglu Y, Desrichard A, Havel JJ, Dalin MG, Riaz N, et al. The head and neck cancer immune landscape and its immunotherapeutic implications. *JCI Insight*. 2016;1(17):e89829.
7. Davoli T, Uno H, Wooten EC, Elledge SJ. Tumor aneuploidy correlates with markers of immune evasion and with reduced response to immunotherapy. *Science*. 2017;355(6322).
8. Taylor AM, Shih J, Ha G, Gao GF, Zhang X, Berger AC, et al. Genomic and Functional Approaches to Understanding Cancer Aneuploidy. *Cancer Cell*. 2018;33(4):676-89 e3.
9. William WN, Jr., Zhao X, Bianchi JJ, Lin HY, Cheng P, Lee JJ, et al. Immune evasion in HPV(-) head and neck precancer-cancer transition is driven by an aneuploid switch involving chromosome 9p loss. *Proc Natl Acad Sci U S A*. 2021;118(19).
10. Nauta IH, Heideman DAM, Brink A, van der Steen B, Bloemena E, Koljenovic S, et al. The unveiled reality of human papillomavirus as risk factor for oral cavity squamous cell carcinoma. *Int J Cancer*. 2021;149(2):420-30.

11. Muijlwijk T, Nijenhuis D, Ganzevles SH, Brink A, Ke C, Fass JN, et al. Comparative analysis of immune infiltrates in head and neck cancers across anatomical sites. *J Immunother Cancer*. 2024;12(1).

REVIEWERS' COMMENTS

Reviewer #1 (Remarks to the Author):

I have reviewed the manuscript and response to previous comments. The authors have appropriately addressed my comments.

Reviewer #2 (Remarks to the Author):

Muijlwijk et al.

“Hallmarks of a Genomically Distinct Subclass of Head and Neck Cancer ”

The revised manuscript is improved in many respects. Some misunderstandings remain regarding central issues surrounding genomic novelty and comparison to HPV-positive HNSCC.

1. The novelty of the approach in increasing rigor by eliminating stromal contaminants, evaluating CNAs in tumors of a larger cohort, solid agreement with TCGA findings and spatial immunological analysis is true and was acknowledged as significant and valuable. As stated in the initial review, what is lacking in novelty is the genomic aspect in terms of identifying new drivers aside from the promoter mutations in TERT and RAC1 that are distinct for the HPV-negative quiet cohort. The lack of TP53, along with the abundance of HRAS, PIK3CA and CASP8 mutations (latter of which the authors point to a potential marker for the quiet subtype) in this group were first noted in detail in the TCGA analysis (Reference 8) and other molecular subtyping studies. While the authors acknowledge this and speculate on the importance of mutated CASP8 and TERT function in cancer, new potential molecular drivers unique to the quiet HPV-negative subtype are not identified. As such, this leaves the potential underlying oncogenic mechanisms and associated genomic signature for the CNA-quiet subgroup as the same or at least partially shared with the CNA-other group, in spite of the quantified differences in clinical, histological and immunological aspects.

2. A note/sentence in the Discussion about the limitation of the study being restricted to just two cohorts in the Netherlands should be included to add proper geographic and demographic perspective to the study.

3. A. The authors are missing the point of the initial critique. From the follow up analysis, late-stage CNA-quiet HPV-negative needs less chemotherapy after surgery than HPV-negative CNA-other. Since HPV-positive patients from the study centers appear to not to have surgical intervention as per past studies from this group, this seems to prove the point that CNA-quiet tumors in the OC may potentially be treated with the same clinical course as HPV-positive OP. Point being- do the favorable clinical and histological characteristics (now matched with more benign treatment) point to a common, low-CNA HNSCC regardless of etiology and anatomical site? Contrary to being “confusing”, survival analysis between HPV-positive OP and the low-CNA HPV-negative cohort could provide support for this idea since these diseases arise from contiguous oral epithelium, and the group has access to these data.

The authors also overstate that HPV-positive OP is only treated with chemotherapy. This may be true in their facility, but it is not true on a global scale. Recent phase II trials in the U.S. have shown that transoral robotic surgery with low-dose radiation on HPV-positive patients is of greater benefit than chemotherapy or radiation alone (PMID:34699271). So much so that this technique is now routinely used as an additional treatment modality for this disease in many such-equipped facilities. As such, HPV-negative CNA-quiet patients might justifiably benefit from this less aggressive treatment. The statement that OP patients are treated with “definitive chemoradiotherapy instead of upfront surgery”, while seemingly true with this cohort, needs to be modified and softened at the least to reflect the ongoing advances in treating HPV-positive disease, especially in the context of potential clinical similarities to HPV-negative CNA-quiet patients.

3. B-D. The rebuttal to these initial critiques has been satisfactorily addressed, but a mention that no patients were treated with cetuximab should be considered, given its initial popularity in combining with radiotherapy in the past decades for all HPV-negative disease.

4. The predominant activating PIK3CA mutational status in the CAN-quiet vs CAN-other again underscores the potential that these tumors may behave in a similar manner to HPV-positive OP, where high numbers of such patients also harbor PIK3CA activating mutations. This fact can be woven into the discussion along with the mention of potential de-escalation aspects noted above (which the authors has already shown is practiced in the clinic in figure R13). Consideration that PIK3CA may be an important convergent node common to these specific disease types should also be highlighted.

5. These critiques have been satisfactorily addressed.

REVIEWER COMMENTS

Reviewer #1:

I have reviewed the manuscript and response to previous comments. The authors have appropriately addressed my comments.

- *We thank the reviewer for the time and effort in reviewing the manuscript. The manuscript has significantly improved due to the insightful comments of the reviewers. The comment regarding genetic heterogeneity was well taken, and we provided additional data on multiple biopsies per tumor to demonstrate that only minor CNA heterogeneity was observed and the definition of CNA-quiet was not impacted by multiregion sequencing.*

Reviewer #2:

The revised manuscript is improved in many respects. Some misunderstandings remain regarding central issues surrounding genomic novelty and comparison to HPV-positive HNSCC.

- *We thank the reviewer for the kind words, we agree that the manuscript has been improved due to the very useful comments of both reviewers.*
1. The novelty of the approach in increasing rigor by eliminating stromal contaminants, evaluating CNAs in tumors of a larger cohort, solid agreement with TCGA findings and spatial immunological analysis is true and was acknowledged as significant and valuable. As stated in the initial review, what is lacking in novelty is the genomic aspect in terms of identifying new drivers aside from the promoter mutations in TERT and RAC1 that are distinct for the HPV-negative quiet cohort. The lack of TP53, along with the abundance of HRAS, PIK3CA and CASP8 mutations (latter of which the authors point to a potential marker for the quiet subtype) in this group were first noted in detail in the TCGA analysis (Reference 8) and other molecular subtyping studies. While the authors acknowledge this and speculate on the importance of mutated CASP8 and TERT function in cancer, new potential molecular drivers unique to the quiet HPV-negative subtype are not identified. As such, this leaves the potential underlying oncogenic mechanisms and associated genomic signature for the CNA-quiet subgroup as the same or at least partially shared with the CNA-other group, in spite of the quantified differences in clinical, histological and immunological aspects.
 - *We believe that we introduce sufficient novelty in the manuscript. We introduce an important group of tumors, define the criteria to identify them for future research, provide quantitative measures by eliminating stromal contaminants, explore clinical and histopathological features, their anatomical location, spatial immunological analysis, and intratumor genetic heterogeneity. In our view this is a very important paper setting the stage for this remarkable CNA-quiet subgroup of tumors.*
 2. A note/sentence in the Discussion about the limitation of the study being restricted to just two cohorts in the Netherlands should be included to add proper geographic and demographic perspective to the study.
 - *The following underlined text has been added to the Discussion of the revised manuscript:*

“CNA-quiet OCSCC typically occur in non-smoking older women. Given the older age at which CNA-quiet OCSCC are diagnosed (70 years compared to 64 years for CNA-other) this subclass - now found to constitute 9.1% of the OCSCC - may well increase within aging populations. Note that this proportion is based on two cohorts in the Netherlands. It is unclear whether another etiological factor is in play. HPV does not play a role in oral cancer and can be excluded (46). As we only had access to FFPE samples and did not perform whole exome sequencing (WES), we could not analyze mutational signatures that might point to a certain etiological factor, reported by Alexandrov et al. (47). Specifically, these signatures are based on mutational patterns likely associated to biological processes or etiological factors, with C>A transversions being linked to smoking, and C>T transitions to ultraviolet light (UV) and age (47). We hypothesize that CNA-quiet OCSCC might fall more into the mutational signature associated with age whereas CNA-others might be more linked to the smoking signature.”

3. A. The authors are missing the point of the initial critique. From the follow up analysis, late-stage CNA-quiet HPV-negative needs less chemotherapy after surgery than HPV-negative CNA-other. Since HPV-positive patients from the study centers appear to not to have surgical intervention as per past studies from this group, this seems to prove the point that CNA-quiet tumors in the OC may potentially be treated with the same clinical course as HPV-positive OP. Point being- do the favorable clinical and histological characteristics (now matched with more benign treatment) point to a common, low-CNA HNSCC regardless of etiology and anatomical site? Contrary to being “confusing”, survival analysis between HPV-positive OP and the low-CNA HPV-negative cohort could provide support for this idea since these diseases arise from contiguous oral epithelium, and the group has access to these data.

The authors also overstate that HPV-positive OP is only treated with chemotherapy. This may be true in their facility, but it is not true on a global scale. Recent phase II trials in the U.S. have shown that transoral robotic surgery with low-dose radiation on HPV-positive patients is of greater benefit than chemotherapy or radiation alone (PMID:34699271). So much so that this technique is now routinely used as an additional treatment modality for this disease in many such-equipped facilities. As such, HPV-negative CNA-quiet patients might justifiably benefit from this less aggressive treatment. The statement that OP patients are treated with “definitive chemoradiotherapy instead of upfront surgery”, while seemingly true with this cohort, needs to be modified and softened at the least to reflect the ongoing advances in treating HPV-positive disease, especially in the context of potential clinical similarities to HPV-negative CNA-quiet patients.

- *Treatment decisions for head and neck cancer are mainly determined by stage and site. Surgery is the upfront mainstay of treatment for OCSCC, and depending on stage, histological findings in the surgical specimen, and the general condition of the patient postoperative radiotherapy or postoperative chemoradiotherapy is applied. Generally OPSCC is treated with definitive radiotherapy or chemoradiotherapy, but the reviewer is correct that other treatments are applied, and we also apply transoral robotic surgery (TORS) at our center. We also agree with the reviewer that at other centers treatment choices may be somewhat different for OPSCC. Of note, this statement on definitive (chem)radiotherapy was only made in the previous point-to-point response, and we did not mention this in the manuscript.*

- *In our view, surgery will remain the primary treatment for OCSCC, and potential de-intensifying schedules for CNA-quiet tumors, should be based on upfront surgery. However, de-escalating treatment, in terms of reducing surgical or post-operative interventions for CNA-quiet OCSCC could certainly be considered. The suggestion that TORS or reduced postoperative radiotherapy schedules might be an option for CNA-quiet tumors is a challenging idea, but this should not be based on prognostic associations with HPV-positive OPSCC, which is in our view a separate disease from both a clinical and biological perspective. HPV targets homology directed repair causing a high sensitivity for cisplatin, HPV-positive cell lines are very sensitive to radiotherapy, field cancerization does not occur with HPV tumors, and the statement of the reviewer that the epithelium is contiguous is not correct. It has been shown that the epithelium in which HPV tumors arise is different, underlying the transforming infection of HPV in tonsillar crypt epithelium. These are all relevant factors explaining the different behavior and favorable treatment outcome for HPV-induced OPSCC, but whether these are as relevant for CNA-quiet OCSCC tumors is unknown. Notwithstanding, we do acknowledge the idea of the reviewer for de-escalation treatment protocols, and we revised the discussion as follows:*

“HPV-negative CNA-quiet OCSCC and HPV-positive HNSCC have some shared clinical (improved prognosis, generally non-smoking) and histological (well-differentiated) characteristics. However, HPV-positive HNSCC mostly occur in the tonsillar crypt epithelium in the oropharynx. While we acknowledge some similarities, HPV-positive OPSCC is considered as a separate disease entity when compared to HPV-negative OPSCC, both at the clinical and biological level. Within OCSCC, HPV rarely occurs and does not play a prognostic role, as reported by Nauta et al. (33). We analyzed the survival of HPV-positive OCSCC, CNA-other and CNA-quiet OCSCC and only CNA-quiet OCSCC showed a more favorable outcome as expected (Supplementary Fig. 13c). Taken together, while for OPSCC the difference between HPV-negative and HPV-positive is critical, for OCSCC, the distinction between CNA-quiet and CNA-other is crucial. Whether treatment de-escalation strategies, such as reduced radiotherapy schedules as applied for HPV-positive OPSCC (34), might become applicable to CNA-quiet tumors, remains to be determined.”

B-D. The rebuttal to these initial critiques has been satisfactorily addressed, but a mention that no patients were treated with cetuximab should be considered, given its initial popularity in combining with radiotherapy in the past decades for all HPV-negative disease.

- *The following underlined text has been added to the legend of table 1:*

“Abbreviations: OCSCC, oral cavity squamous cell carcinoma; T, N and disease stage according to TNM classification 8th edition, 2017. Pathological stage was used when available; when the patient was not treated with surgery, clinical stage was used. Pattern of invasion (POI) as reported by Heerema et al. (18). For treatment type, only advanced disease-staged (III-IV) tumors with curative intent were included. No patients were treated with cetuximab. ^a p-value obtained by Mann-Whitney test, ^b p-value obtained by Chi-squared test. Unknowns were excluded from the analysis.”

4. The predominant activating PIK3CA mutational status in the CNA-quiet vs CNA-other again underscores the potential that these tumors may behave in a similar manner to HPV-positive OP, where high numbers of such patients also harbor PIK3CA activating mutations. This fact

can be woven into the discussion along with the mention of potential de-escalation aspects noted above (which the authors has already shown is practiced in the clinic in figure R13). Consideration that PIK3CA may be an important convergent node common to these specific disease types should also be highlighted.

- *In the Discussion, we compare CNA-quiet with HPV-positive OPSCC and suggestions for treatment de-escalation (see point 3). We do, however, not mention PIK3CA in this comparison with HPV-positive OPSCC, as it is not significantly enriched in CNA-quiet versus CNA-other (adjusted p-value=0.13). Hence, the data do not support PIK3CA as an important node in CNA-quiet OCSCC.*

5. These critiques have been satisfactorily addressed.